# Evidence for multi-fragmentation and mass shedding of boulders on rubble-pile binary asteroid system (65803) Didymos

Asteroids smaller than 10 km are thought to be rubble piles formed from the reaccumulation of fragments produced in the catastrophic disruption of parent bodies. Ground-based observations reveal that some of these asteroids are today binary systems, in which a smaller secondary orbits a larger primary asteroid. However, how these asteroids became binary systems remains unclear. Here, we report the analysis of boulders on the surface of the stony asteroid (65803) Didymos and its moonlet, Dimorphos, from data collected by the NASA DART mission. The size-frequency distribution of boulders larger than 5 m on Dimorphos and larger than 22.8 m on Didymos confirms that both asteroids are piles of fragments produced in the catastrophic disruption of their progenitors. Dimorphos boulders smaller than 5 m have size best-fit by a Weibull distribution, which we attribute to a multi-phase fragmentation process either occurring during coalescence or during surface evolution. The density per $km^2$ of Dimorphos boulders ≥1 m is 2.3x with respect to the one obtained for (101955) Bennu, while it is 3.0x with respect to (162173) Ryugu. Such values increase once Dimorphos boulders ≥5 m are compared with Bennu (3.5x), Ryugu (3.9x) and (25143) Itokawa (5.1x). This is of interest in the context of asteroid studies because it means that contrarily to the single bodies visited so far, binary systems might be affected by subsequential fragmentation processes that largely increase their block density per $km^2$. Direct comparison between the surface distribution and shapes of the boulders on Didymos and Dimorphos suggest that the latter inherited its material from the former. This finding supports the hypothesis that some asteroid binary systems form through the spin up and mass shedding of a fraction of the primary asteroid.

The analyses of Main Belt (MB) and Near-Earth Asteroids' (NEA) shapes and spins[1] coupled with the impact physics' numerical models[2,3] have suggested that the majority of asteroids with sizes between ~0.2 and 10 km are rubble piles[4,5]. This means that they are non-monolithic aggregates made of numerous boulders inherited from catastrophically disrupted parent bodies, which later coalesced under the influence of gravity. In this context, the analysis of blocks and their size frequency distribution (SFD) enable us to investigate the evolution and collisional history of the parent bodies and shed light on the geomorphological processes shaping the rubble piles surfaces[6–8]. We hereafter use blocks and boulders as synonyms, being positive reliefs[6,7] larger than 0.25 m, which seem to protrude from the ground where they stand, and detectable in different, increasingly higher spatial scale images with the constant presence of an elongated shadow (see Methods).

Images of NEAs acquired by space missions enabled the most detailed boulder analyses ever obtained on small bodies[9–14]. In

✉ e-mail: maurizio.pajola@inaf.it

particular, the investigation of block SFDs obtained from size range between few centimeters to hundreds of meters, were found to commonly follow power-law fits[9–11,13–15]. From a formative perspective, this means that these boulders have been generated by a sudden fragmentation, as an impact event, and leading to a distribution of remnants characterized by fractals[8,14]. Such a result, coupled with both the extremely blocky nature of the observed surfaces and the presence of the largest boulders that are usually ~1/10 the NEAs' diameters, support an impact scenario for the boulders, whether from craters' emplacement or from the progenitor catastrophic disruption[8,16].

The power-law indexes obtained on global counts for C-complex asteroids (henceforth called *carbonaceous* asteroids) (162173) Ryugu and (101955) Bennu are $-2.65 \pm 0.05$ for boulders with sizes 5–160 m (hereafter, all size-ranges indicated show the maximum boulder size identified for each mentioned NEA) and $-2.5 \pm 0.1$ for boulders with sizes 0.2–100 m, respectively. These values indicate that such bodies are rubble piles dominated by impact processes[13,14,17]. On the other hand, the global boulder distribution of the S-complex (henceforth called *stony*) NEA (25143) Itokawa has a power-law index of $-3.05 \pm 0.14$ for boulders 5–50 m, although on specific areas of the asteroid the boulder SFD was fit by broken power-law curves[15]. Stony NEAs (433) Eros and (4179) Toutatis have a SFD with a power-law index of $-3.25 \pm 0.14$ for boulders 10–140 m and of $-4.4 \pm 0.1$ for boulders 20–61 m, respectively[9,11,15,16]. As for the carbonaceous counterpart, such trends are reflective of a rubble pile origin as a result of a progenitor catastrophic disruption (with Eros being a possible single-shard fractured exception, later modified by multiple impacts[18,19]). Nevertheless, the power-law index is higher on stony asteroids than on carbonaceous asteroids. This suggests that on the former, more of their surface mass is contained in smaller boulders rather than on the latter[14], hence reflecting how distinct materials respond differently to meteoroid bombardment and thermal cracking[20].

In this work, we analyze the boulder SFD derived from the surface of the stony/Sq-type NEA binary (65803) Didymos[21,22], which was the target of the NASA Double Asteroid Redirection Test (DART) spacecraft[23]. On 26 September 2022, this mission intentionally impacted Dimorphos, the natural satellite of Didymos, successfully demonstrating the kinetic redirection technique for planetary defense purposes[24,25]. The boulder SFD of the system, obtained through DART high-resolution images, is therefore a powerful metric for distinguishing among previously proposed formation scenarios of binary asteroids[26–29].

The Didymos system, with a primary of size[30] $819 \times 801 \times 607$ m and a secondary of size[31] $177 \times 174 \times 116$ m, belongs to the largest group of NEA binaries with secondary/primary size ratios[32] of $0.1 \leq \frac{secondary\ size}{primary\ size} \leq 0.6$. One formation hypothesis of such binary bodies is that due to the Yarkovsky–O'Keefe–Radzievskii–Paddack (YORP)[33] effect, a larger primary might have experienced continuous spin-up to reach its spin limit. As a consequence, a mass shedding event or fission of some fraction of its body[26,34,35] occurred. Ejected materials from the primary are predicted to remain in orbit within the system and reaccumulate outside the Roche limit into a small satellite. If the formation of Dimorphos is related to the top shape and rapid spin-up of Didymos by the YORP effect[26,34,36,37], it is expected that its boulders previously belonged to the equatorial region of the primary and have a comparable inherited SFD. Moreover, if this interpretation holds true, we could also expect some sort of equatorial block depletion on Didymos, as a result of the spin-up process, followed by the mass-shedding event.

Here we quantitatively test this hypothesis by evaluating if the boulder SFD on Didymos and Dimorphos can be fit by either a power-law or a different curve. This fit, coupled with the identification of the boulder maximum sizes, the block number density per unit area, and the study of the global properties of the identified boulders, provide insights into the different formative and degradation processes of the binary system as a whole.

## Results

During the last 5 min of the DART mission, the Didymos Reconnaissance and Asteroid Camera for OpNav (DRACO) scientific camera[38] imaged the surfaces of the binary system over a range of spatial scales[31], with a constant phase angle of $\sim 59°$. Such phase angle falls in the 40–80° range which is particularly good to identify protruding surface features as boulders[13,14,39,40], because they all show the presence of a well-defined shadow, which ease their identification. To perform the SFD analysis, we used the DRACO images with the best available resolution that fully covered the visible, illuminated and well-contrasted surface of each asteroid, with the presence of features characterized by crisp boundaries. The two images used to identify Dimorphos boulders were acquired at a distance of 52.56 km and 40.73 km, corresponding to spatial scales of 0.26 and 0.20 m/pixel, respectively. For Didymos the four images analyzed were taken at distances of 990–633 km with spatial scales ranging from 4.9 to 3.3 m/pixel. Using well-established image processing approaches, see refs. 39,40 and references herein, such pixel scales allow the identification of all boulders in the visible and lit terrain that are larger than 0.6–0.8 m on Dimorphos and larger than 10–15 m on Didymos. We highlight that the resolution of the images on both bodies is largely different. Nevertheless, what is important for the origin and degradation implications of the two bodies is the boulder SFD trends, even at different size-ranges, and the resulting fitting curve indices[8,10,11,13,14]. Using the Small Body Mapping Tool software (SBMT[41]), we directly projected the DRACO images onto the latest available shape models of Dimorphos and Didymos (v004[42]), manually fitting the boulders as ellipses (see Methods). The value of each ellipse's major axis was then used as the maximum diameter of the corresponding boulder, assuming that each boulder's maximum extents are exposed on the surface. For each boulder we derived its apparent axial ratio, i.e., the ratio between the semi-minor axis and the semi-major axis of the ellipse. Moreover, we determined for the center of each boulder the gravitational slope (°), the gravitational acceleration (m/s²), and the gravitational potential (J/kg), taking into account both Dimorphos's rotation as well as Didymos' tides. In addition, we derived the orientation (°) of the boulder, which is the angle between the ellipse major axis and the longitude line it lies on as projected onto the surface[30].

### Dimorphos

On Dimorphos, we identified a study area of 0.0132 km², outside of which all surface features appear distorted and stretched due to high emission angles (Fig. 1A). Within this area we counted 4734 boulders (Fig. 1B), finding a maximum size of 16 m. The mode, median and mean of the boulder distribution are 0.7 m, 1.1 m, and 1.4 m, respectively.

To be conservative on the boulder counts, and avoid any possible boulder size misinterpretation which often happens at the smallest dimensions[14], we decided to increase the three-pixel sampling rule to five pixels, i.e., setting a lower size limit of a boulder that is 1.0 m in size to our data.

The cumulative number of boulders per km² versus size (in meters) is represented in the log-log graph of Fig. 1C. The number density/km² of boulders $\geq 1.0$ m is ~203,000, for blocks $\geq 5.0$ m is 6300, while it is 833 for boulders larger than 10.0 m. To evaluate if the resulting SFD can be fit by a power-law curve or not, we made use of the ref. 43 methodology (see Methods), which returns a scaling parameter $\alpha$ (also called power-law index) of $-2.5 \pm 0.2$ and a completeness limit $x_{min}$ of $2.0 \pm 0.5$ m. Considering the significance level of 0.1[43], the p-value we obtained, computed from 2500 Kolmogorov-Smirnoff statistical tests, is <0.1. This result suggests that a single power-law distribution is not a good fit for the boulders SFD $\geq 2$ m. Nevertheless,

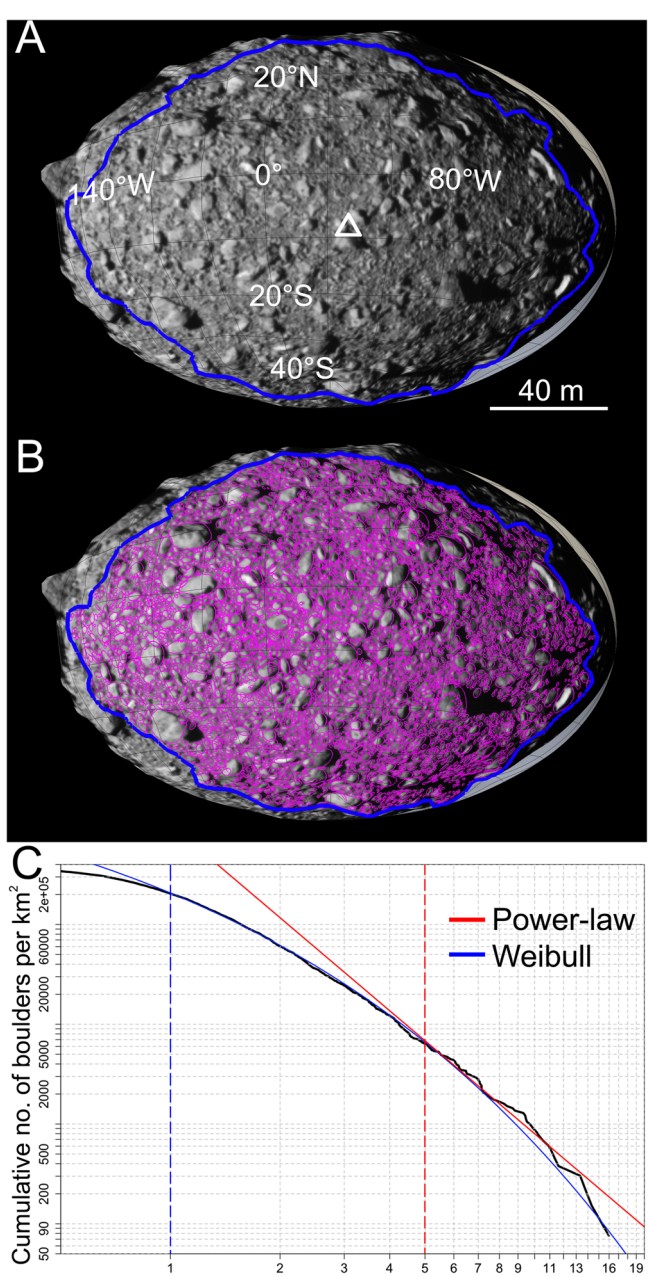

**Fig. 1 | Dimorphos boulders and size-frequency distribution. A** The study area, highlighted with a blue polygon, identified on Dimorphos. The white triangle represents the DART impact location. **B** The 4734 boulders identified on Dimorphos outlined in pink. **C** The Dimorphos cumulative number of boulders per km². The derived power-law fitting curve in red is obtained for boulders ≥5.0 m. The Weibull curve, highlighted in blue, is obtained for boulders ≥1.0 m. Source data are provided as a Source Data file.

to compare our SFD with power-law fitting curves available from the NEA literature, we attempted to identify which is the minimum size ($x_{min}$) that the ref. 43 technique identifies and for which it returns a valid power-law fit. We found that our distribution is well fit by a power-law curve (p-value of 0.19) when an $x_{min}$ of 5.0 ± 1.2 m is identified, returning an $\alpha$ value of −3.4 ± 1.3.

Even if the −3.4 index represents the Dimorphos ≥5 m SFD, it is clear (Fig. 1C) that at smaller sizes (1–5 m), a departure from the power-law curve is real and not a resolution effect. For the widest possible boulder size range (1–16 m, Fig. 1C), we find that the best fitting curve to the data is the Weibull function (see Methods), which is in the form

of $y = A\,exp^{-(\frac{x}{\lambda})^k}$, with the derived parameters of $\lambda = 0.0327$, $k = 0.39589$ and $A = 9995083.23$.

In Fig. 2A–F, we plot the boulder size versus latitude, longitude, gravitational slope, gravitational acceleration, potential, and the heliocentric average insolation[44]. In Fig. 3A–C, we show the apparent axial ratio for all Dimorphos boulders ≥1.0 m (mean value of 0.66 with a $\sigma$ of 0.20), ≥3.0 m (mean value of 0.56 with a $\sigma$ of 0.19) and ≥5.0 m (mean value of 0.53 with a $\sigma$ of 0.20).

### Didymos
On Didymos, we identified a study area that is 0.3660 km² wide, within which no surface features appear distorted and stretched (Fig. 4A). We identified a total number of 169 boulders (Fig. 4B), getting a maximum size of 93 m. This total number is smaller than the one we identified on Dimorphos due to the much lower spatial resolution of the DRACO images of Didymos, however, it still allows to derive an SFD and related statistics, even at different size-ranges with respect to the secondary (Fig. 4C). The mode, median and mean of the boulder distribution are 13.9 m, 21.8 m, and 23.7 m, respectively. As for Dimorphos, to be conservative on the boulder counts and avoid any possible boulder size misinterpretation, we decided to increase the three-pixel sampling rule to five-pixels, i.e., setting a lower size limit of a boulder that is ~16.5 m in size to our data. The number density/km² is 353 for ≥16.5-m boulders, 273 for ≥20.0-m boulders, 74 for ≥30.0-m boulders, 30 for ≥40.0-m boulders and 14 for ≥50.0-m boulders. As for Dimorphos, we used the ref. 43 methodology to evaluate if a power-law curve can fit the resulting SFD (see Methods). From Fig. 4C the cumulative number of boulders per km² is well fit by a power-law curve with $\alpha = -3.6 \pm 0.7$ and $x_{min} = 22.8 \pm 2.3$ m. The p value derived from 2500 Kolmogorov-Smirnoff statistical tests is 0.6, i.e., well above the 0.1 significance level. Such result shows that despite a lower spatial scale of the DRACO imagery dataset for Didymos, a boulder SFD with a significative fit and associated index is derived.

As for Dimorphos, we plot in Fig. 5A–E the boulder size versus latitude, longitude, gravitational slope, gravitational acceleration, and potential. In Fig. 6A–C, we show the apparent axial ratio for all Didymos boulders ≥16.5 m (mean value of 0.86 with a $\sigma$ of 0.17), ≥20.0 m (mean value of 0.82 with a $\sigma$ of 0.17) and for boulders ≥30.0 m (mean value of 0.72 with a $\sigma$ of 0.19).

## Discussion
### Evidence for catastrophic disruption of the parent body
The previously studied global boulder SFDs derived from NEAs (Fig. 7A) have been generally fit by single power-law curves. Nevertheless, recent works by ref. 45 on Ryugu, and by ref. 14 on Bennu, started to make use of the Weibull function in order to fit their boulder SFD. In the first case, ref. 45 (Fig. 2) showed that the power-law fit is a good match for boulders larger than 1–2 m, but if the full cm- to decameter-size range is considered, then the particle SFD is better described by the Weibull curve, especially for boulder-cobbles with sizes below 1 m. On Bennu, ref. 14 applied the same Weibull fit of Ryugu (Fig. 15), showing that this could also be suitable to represent the cm- to decameter-size range SFD. Nevertheless, such curve indicated some under-estimation for sizes >20 m, while it partially over-estimated the boulders in the 0.5–8.0 m size range. On the contrary, the power-law fit showed to be better representative of the overall SFD, as is the case for all remaining NEAs Eros[9], Toutatis[11], and Itokawa[10] (for the latter multiple power-law fitting curves[15] have been introduced to explain the different boulder trends observed on the surface). From a formation perspective, a single power-law SFD indicates a single-event fragmentation that leads to a branching tree of cracks with a fractal character[46–48]. Specifically, a power-law fit generally supports impact-generated boulders and ejecta, whether from a crater-forming event or from a parent body catastrophic disruption[8,10,11,14,16]. As shown in Fig. 7B, boulders on all stony NEAs previously visited by spacecrafts−

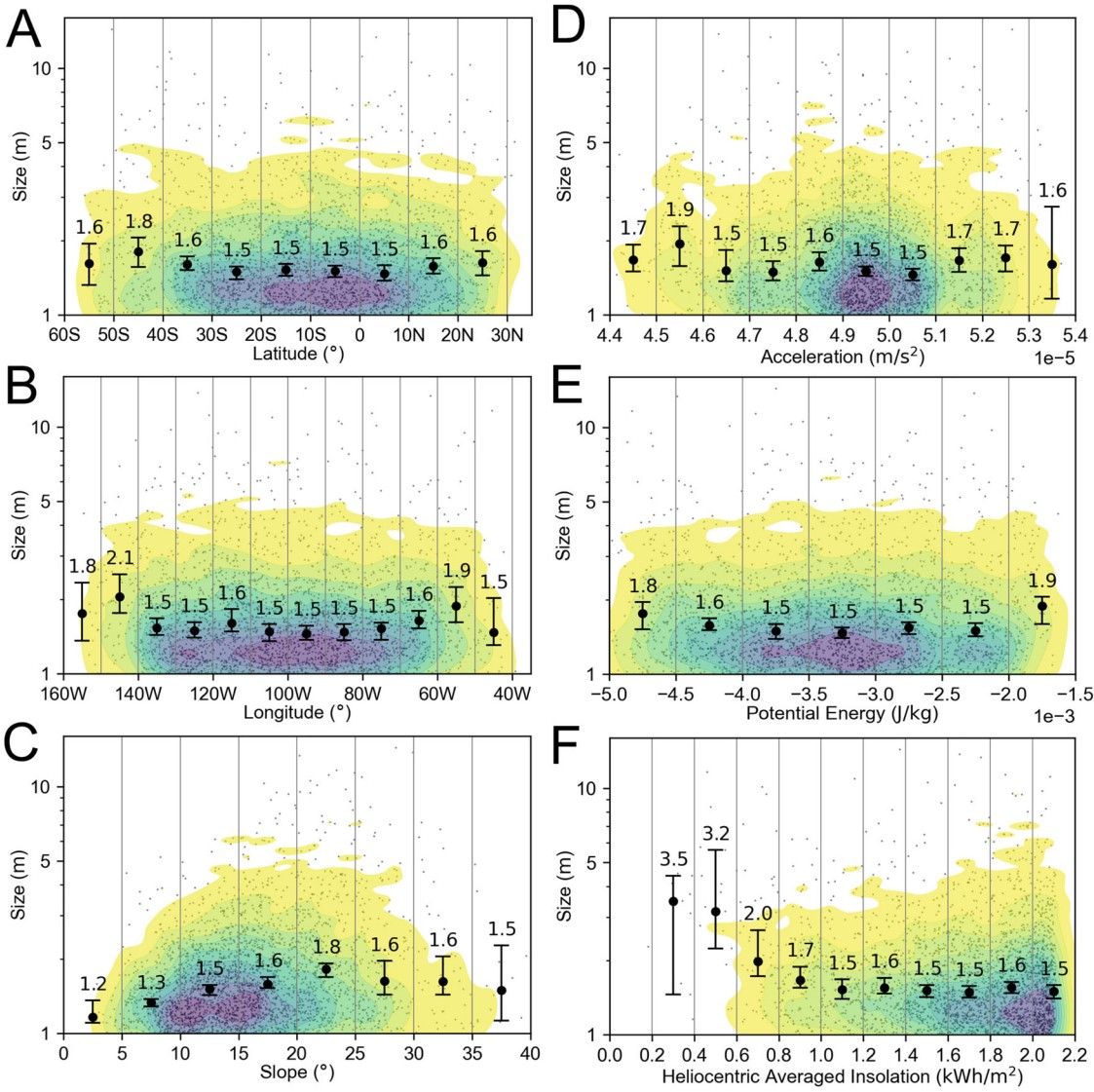

**Fig. 2 | Dimorphos boulders' global distributions.** The Dimorphos boulder size versus latitude (**A**), longitude (**B**), gravitational slope (**C**), gravitational acceleration (**D**), potential (**E**), and heliocentric average insolation (**F**). The black dots represent the identified boulders, while the contours depict the bi-variate kernel density estimate (KDE) of their distribution in the specified space. Contours are traced at 10% iso-proportions of the normalized probability density estimate. Error bars display the median boulder size within evenly spaced bins, along with the corresponding 99% two-sided bootstrap confidence interval. We underscore that a detailed discussion of all the presented diagrams and their implications for formative and degradation processes in presented inside the "Discussion−Evidence for formation of Dimorphos via mass shedding" section of the manuscript. Source data are provided as a Source Data file.

Itokawa, Eros, and Toutatis−are characterized by power-law fitting curves with indices steeper than −3.0. In particular, Itokawa shows a power-law index of −3.05 ± 0.14 for boulders ≥5 m[10], Eros has a power-law index of −3.25 ± 0.14 for boulders ≥10 m[9], while Toutatis is characterized by a power-law index of −4.4 ± 0.1 for boulders ≥20 m[11]. On the contrary, on carbonaceous asteroids Ryugu and Bennu the power-law indexes obtained are −2.65 ± 0.05 for boulders ≥5 m[13] and −2.5 ± 0.1 for boulders with sizes ≥0.2 m[14], respectively. Such indices all confirm an impact-related formation that led to a SFD characterized by fractals[46]. However, the variance in trends between them confirms that the power-law index is greater among stony asteroids compared to carbonaceous ones. This underscores the distinct responses of materials (stony versus carbonaceous) to meteoroid impacts and thermal cracking, as previously suggested by ref. 20. The Dimorphos power-law index α of −3.4 ± 1.3, obtained for boulders ≥5 m, and Didymos power-law index α of −3.6 ± 0.7, derived from boulder sizes ≥22.8 m, confirm this generally steeper stony boulder SFD (Fig. 7B) when compared to the carbonaceous one, as well as they indicate an impact-related origin for the identified boulders[10,11,16]. As for the other visited bodies, this evidence, coupled with the maximum identified boulder dimensions (93 m on Didymos and 16 m on Dimorphos) that both exceed 1/10 the NEAs' diameters, imply that such asteroids are collections of debris resulting from the catastrophic breakup of a larger parent body[8,16,30,31], followed by the reaccretion of part of its fragments.

This formation scenario is also supported by the resulting mean axial ratio values found for Dimorphos (Fig. 3A–C) and Didymos boulders (Fig. 6A–C). Indeed, laboratory impact experiments performed with different projectile velocities, target shapes, compositions and strengths have shown that the mean axial ratio (width-to-length) of the resulting fragments is usually distributed around 0.70–0.74[49]. Despite being considerably smaller than surface boulders, the laboratory mm- to cm-size pebbles/cobbles have demonstrated to be extremely useful when their mean axial ratio is compared to the apparent mean axial ratio derived from impact-generated boulders

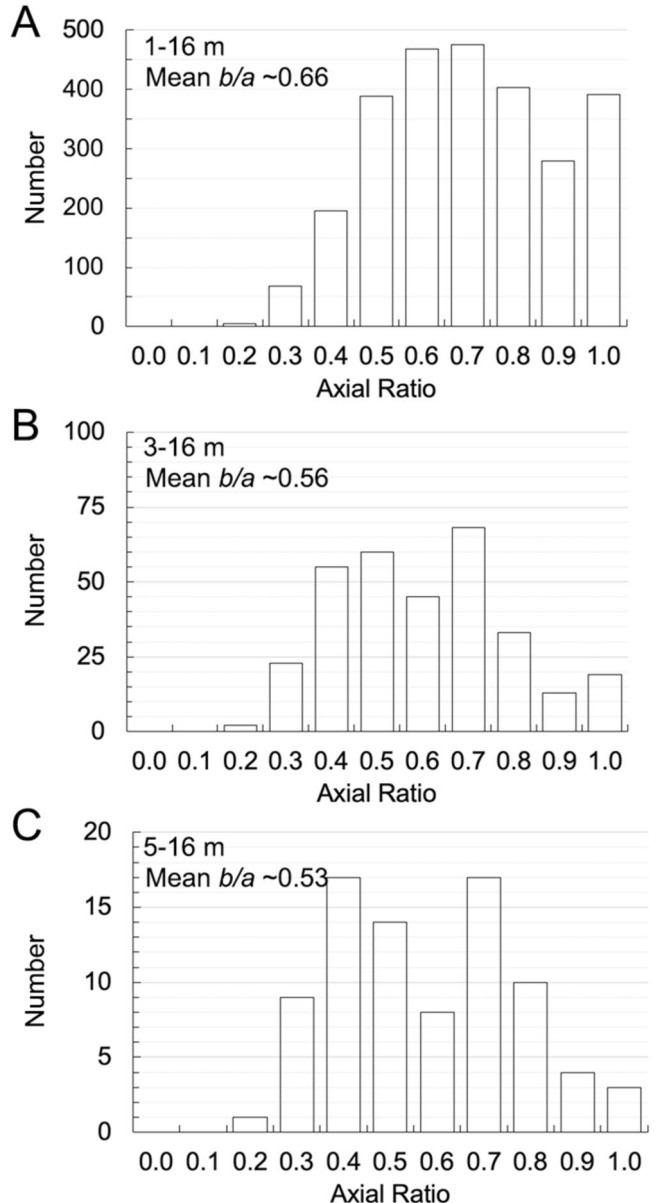

**Fig. 3 | Dimorphos boulders' apparent axial ratio.** The apparent axial ratio for all Dimorphos boulders ≥1.0 m (**A**), ≥3.0 m (**B**) and ≥5.0 m (**C**). Source data are provided as a Source Data file.

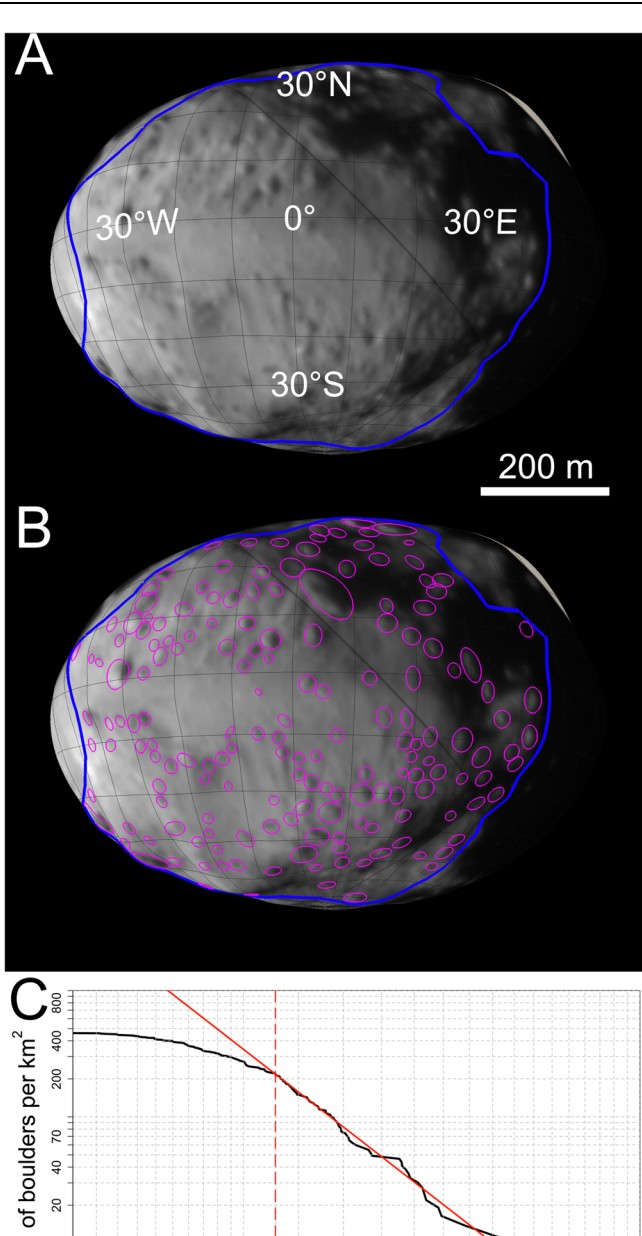

**Fig. 4 | Didymos boulders and size-frequency distribution. A** The surface of Didymos where we identified all boulders. **B** The study area highlighted with a blue polygon. The 169 identified boulders are outlined in pink. **C** The Didymos cumulative number of boulders per km². The derived power-law fitting curve in red, is obtained for boulders ≥22.8 m. Source data are provided as a Source Data file.

observed on NEAs as Eros, Itokawa, and Ryugu[16,49] (we recall that there are no Bennu, nor Toutatis values available for such comparison). For this reason, we opted to conduct a similar comparison using our Didymos and Dimorphos data, as done by refs. 16,49. The axial ratio found on Eros (0.72 for boulders ≥30 m), Ryugu (0.68 for boulders ≥5 m) and Itokawa (0.63 for boulders ≥5 m) suggested that all boulders belonging to these NEAs are still the result of the collisional disruption of their parent bodies and any differences from the 0.70–0.74 range is mainly attributed to the inclination of the boulders laying on the asteroid surfaces[16]. Moreover, ref. 16 suggested that the apparent mean axial ratio of similar-size boulders decreases with asteroid size. If we consider all Dimorphos boulders ≥5.0 m, we get an apparent mean axial ratio of 0.53 with a $\sigma$ of 0.20. This is consistent with the previously mentioned trend—Dimorphos is the smallest NEA ever visited so far and has the smallest mean axial ratio. In addition, the apparent mean axial ratio of Dimorphos boulders increases from 0.56 to 0.66 when considering smaller boulders, as foreseen by ref. 16. Such behavior is

explained by the fact that smaller boulders have a vertical axis that gradually becomes perpendicular to the surface during granular processes[50], owing to their lower friction angle and gravitational stability. Indeed, as mentioned in ref. 16, once the reaccretion process has occurred, smaller boulders are redistributed due to seismic shaking caused by repeated impacts. This is favored by their higher mobility due to the lower friction angle. Such migration consequently affects their orientation, letting them "lying flat" on the surface. This is the reason why the apparent axial ratio of smaller boulders tends to

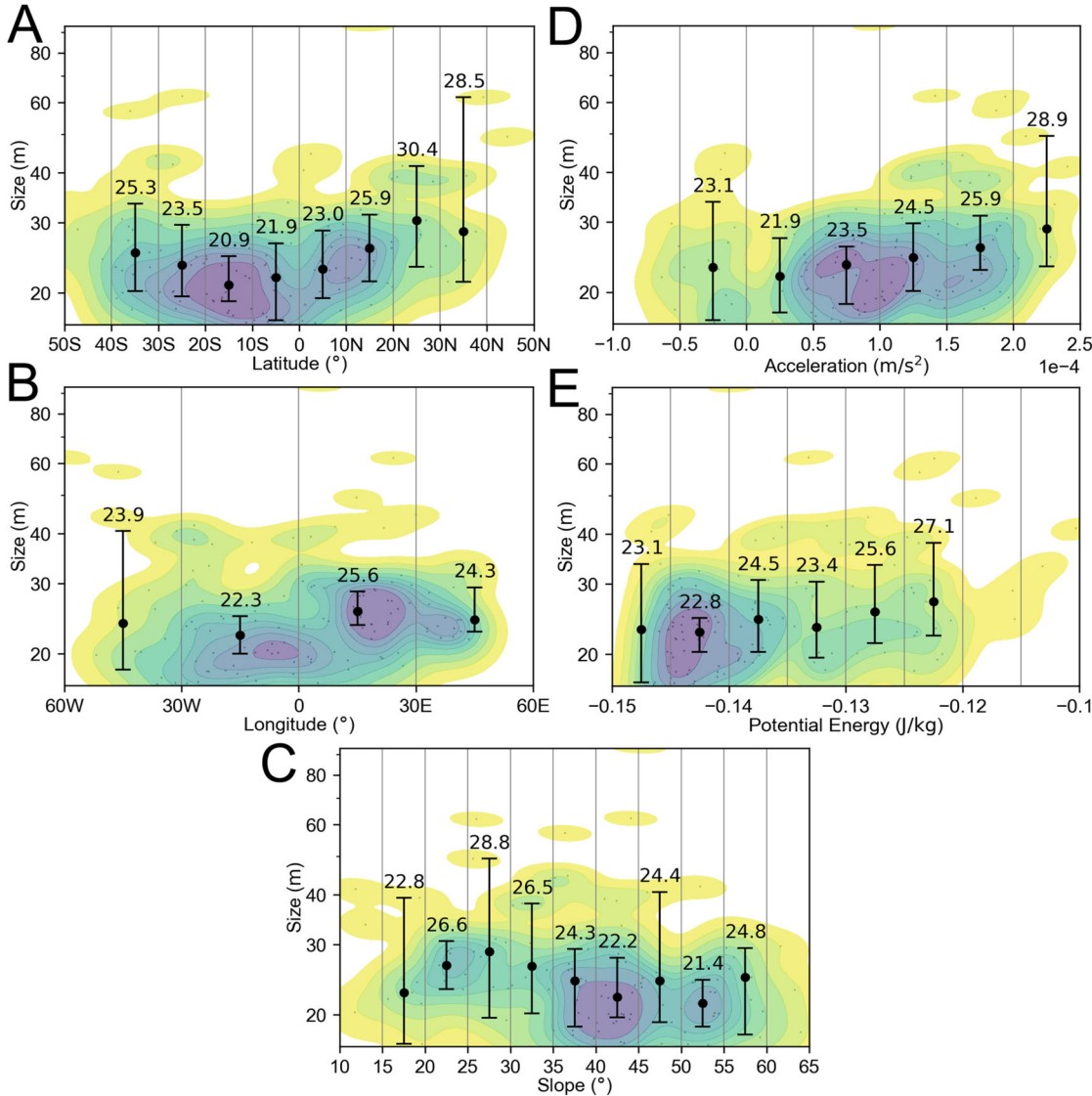

**Fig. 5 | Didymos boulders global distributions.** The Didymos boulder size versus latitude (**A**), longitude (**B**), gravitational slope (**C**), gravitational acceleration (**D**), and potential (**E**). The black dots represent the identified boulders, while the contours depict the bi-variate kernel density estimate (KDE) of their distribution in the specified space. Contours are traced at 10% iso-proportions of the normalized probability density estimate. Error bars display the median boulder size within evenly spaced bins, along with the corresponding 99% two-sided bootstrap confidence interval. We underscore that a detailed discussion of all the presented diagrams and their implications for formative and degradation processes in presented inside the "Discussion–Evidence for formation of Dimorphos via mass shedding" section of the manuscript. Source data are provided as a Source Data file.

approach the one of laboratory fragments as the size of small boulders decreases[16]. The mean axial ratio of 0.86 ($\sigma$ of 0.17) found for Didymos boulders $\geq 16.5$ m means that such body appears to be characterized somehow by less elongated boulder shapes. Nevertheless, if we consider all boulders $\geq 30.0$ m we get an apparent axial ratio of 0.72 and a $\sigma$ of 0.19, which is equal to the one obtained for Eros. The mean axial ratios values found from the analysis are in agreement with laboratory experiments and consistent with the other NEAs, hence indicating that both Didymos and Dimorphos blocks are the result of the catastrophic disruption of their parent body. Nevertheless, if we interpret the mean axial ratio values for boulders $\geq 5.0$ m on Dimorphos, Itokawa, and Ryugu, we clearly notice that on the latter, a much higher surface mobility has occurred making them appear almost parallel to the surface (mean axial ratio of 0.68). On the contrary, Itokawa boulders $\geq 5.0$ m appear in an intermediate situation (0.63), while Dimorphos boulders $\geq 5.0$ m seem to not have been affected by any surface

movement yet, considering that the 0.53 value is the smallest obtained. On the contrary, Eros boulders $\geq 30.0$ m (as for Didymos case) have an apparent axial ratio which is 0.72, hence indicating that they are parallel to the surface.

The finding that the whole SFD of boulders on Dimorphos follows a Weibull distribution suggests that a past or ongoing processes may have decreased the number of meter-size boulders visible on Dimorphos' surface. Indeed, a Weibull distribution is commonly thought to result from sequential fragmentation[51], and it is widely used in fracture theory[46,48,51–53]. In addition, such distribution is often used to describe the particle distribution that is derived from grinding experiments[54], i.e., a multiple-event fragmentation. In this context, the Dimorphos Weibull boulder SFD could be explained by a multi-phase fragmentation process related to the boulder's relative velocities and sizes, which occurred during the accretion of the secondary, as discussed in the section below.

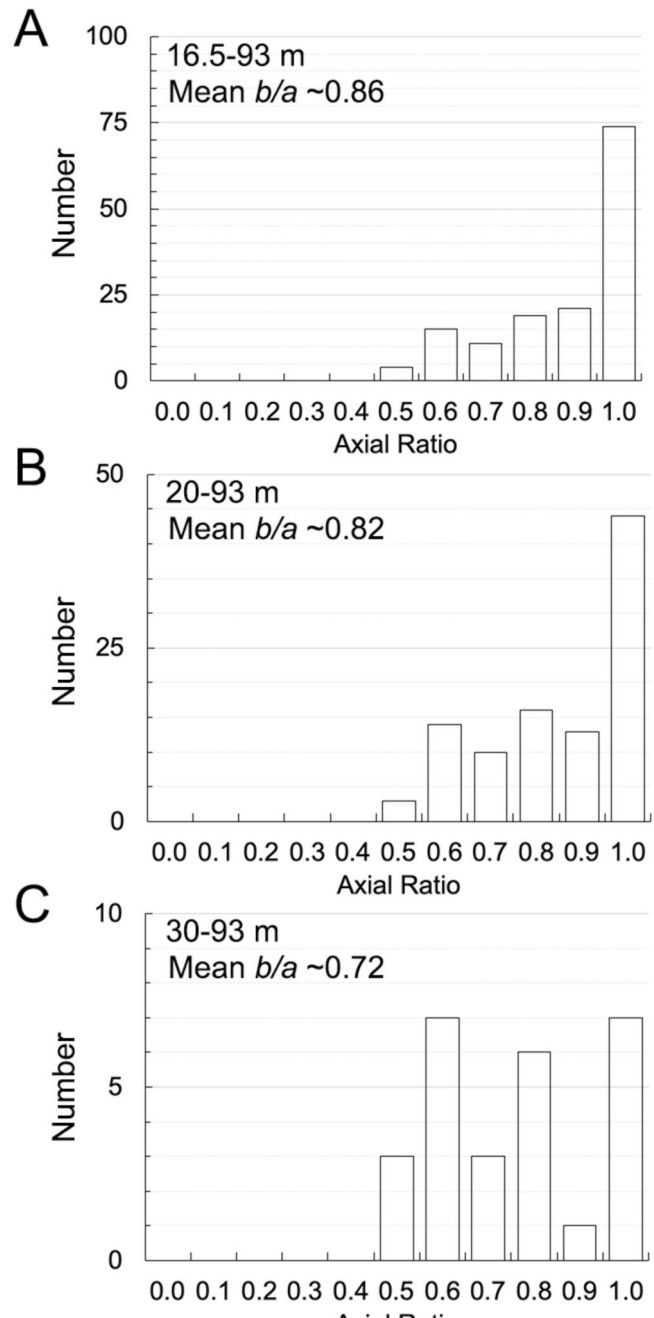

**Fig. 6 | Didymos boulders' apparent axial ratio.** The apparent axial ratio for all Didymos boulders ≥16.5 m (**A**), ≥20.0 m (**B**), and ≥30.0 m (**C**). Source data are provided as a Source Data file.

### Evidence for formation of Dimorphos via mass shedding

Although the fitting ranges for Dimorphos and Didymos are slightly different, i.e., ≤16 m for the secondary, while ≥22.8 m for the primary, the fact that both asteroids' power-law fits reasonably overlap within error bars (Fig. 7) suggests that Dimorphos' surface has directly inherited part of Didymos' boulder SFD. This result supports the findings of dynamical studies[55] which show that Dimorphos could have formed as the result of a mass shedding event from Didymos. Moreover, the occurrence of landslides and mass shedding from the primary while forming the secondary due to rapid spin-up caused by the YORP effect[34,55] is also one of the possible explanations of the resulting Weibull Dimorphos SFD, since they may result in specific particle size-segregation and/or size-depletion. Hence, such processes have a

significant role in shaping the final surface appearance of low-gravity bodies since they can cause the formation of localized deposits characterized by different texture, or they can result in particle SFD which are characterized by multiple curves, i.e., different formative and/or degradation events. In particular, there is evidence that particle size segregation occurs during shallow gravity-driven surface flows and that the larger particles tend to migrate to the top and attain higher speed in a freely flowing granular system, see, e.g., refs. 56,57. This segregation could be due to small particle percolation, size-dependent collisional, and frictional exchange of momentum. The Didymos spin is currently 2.26 h[23]. Nevertheless, as showed by ref. 55 only a slightly shorter period of 2.2596 h could trigger surface landslides and mass shedding from the primary[30]. If segregation works effectively under low-gravity conditions, this would indicate that when Didymos was spinning at its critical limit and surface grains started to move downslope, large boulders tended to acquire larger kinetic energy and slide faster. Because the Coriolis forces acting on the boulders are proportional to the sliding velocity[58], this mechanism would preferentially eject larger boulders from the surface. The segregation effect which occurred during the mass shedding event is a hypothesis we advance, nevertheless, the exact size limit below which there is a general boulder number decrease has not yet been feasibly modeled.

The Dimorphos formation scenario via mass wasting is also supported by plotting the size of its boulders versus latitude, longitude, slope, gravitational acceleration, and potential (Fig. 2A–E). Indeed, we found that the boulders appear to be randomly distributed on the surface. Nevertheless, the clear cut-off of boulders located on gravitational slopes in the 35–45° range suggests that no blocks can remain stable at larger inclinations[30]. Consequently, this implies that the angle of repose, denoting the maximum angle at which granular material can be piled without collapsing, for Dimorphos's material falls within this specific range. Contrarily, on Didymos the size versus latitude plot (Fig. 5A) suggests that the largest sizes are more concentrated at the highest latitudes where the rough highland is located[30], i.e., further from the equatorial triangular-shaped ridge. This supports the interpretation that the equatorial "smooth" lowland of Didymos[30] is characterized by boulders with sizes that are close to or under the DRACO detection limit. This might be the result of the mass-shedding event that generated Dimorphos from Didymos equatorial band[55], which later flattened, being characterized only by small rubbles. The boulder size versus longitude, slope, gravitational acceleration, and potential (Fig. 5B–E) do not exhibit a specific trend indicating a random distribution, as observed for Dimorphos. Nevertheless, a gravitational slope boulder cut off in the 55–65° range suggests that on Didymos there is surface cohesion which is larger than Dimorphos's one.

The alignment of Dimorphos's boulders' semi-major axes relative to the local north could offer insights into their potential source location on Didymos. Indeed, if these boulders exhibit random orientations, it would imply an accumulation of boulders sourced from multiple locations on Didymos. By contrast, a preferential orientation may indicate preferential mass shedding from certain regions on Didymos or significant rearrangement of the surface during Dimorphos evolution. For Dimorphos boulders ≥1 m with an apparent axial ratio <0.9, we find that the orientation distribution is a combination of a smaller family of boulders with a general random orientation and a larger family of boulders with a preferential orientation direction in the 95–140° range. This is independent from the size of the boulder as well as from the local gravitational slope of the asteroid and it peaks around 125° (Fig. 8A). On the contrary, Didymos boulders ≥16.5 m with an apparent axial ratio <0.9 are aligned in a 5–50° direction, peaking around 28° (Fig. 8B). As for Dimorphos, this result is independent from the size of the boulder as well as from the local gravitational slope of the asteroid.

One possible explanation for Didymos blocks' alignment could be a preferential NNE-SSW regolith migration, already noted on Bennu[59]

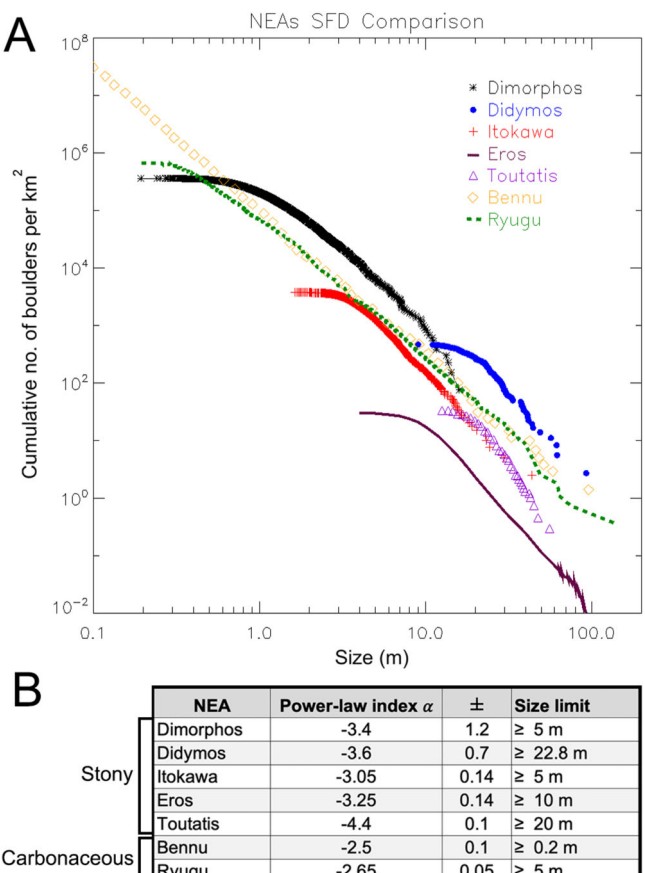

| | NEA | Power-law index $\alpha$ | $\pm$ | Size limit |
|---|---|---|---|---|
| Stony | Dimorphos | -3.4 | 1.2 | $\geq$ 5 m |
| | Didymos | -3.6 | 0.7 | $\geq$ 22.8 m |
| | Itokawa | -3.05 | 0.14 | $\geq$ 5 m |
| | Eros | -3.25 | 0.14 | $\geq$ 10 m |
| | Toutatis | -4.4 | 0.1 | $\geq$ 20 m |
| Carbonaceous | Bennu | -2.5 | 0.1 | $\geq$ 0.2 m |
| | Ryugu | -2.65 | 0.05 | $\geq$ 5 m |

**Fig. 7 | NEAs SFD comparison. A** Comparative plot showing all global boulder SFD obtained for the visited NEAs. **B** The resulting power-law indices and associated error bars obtained in the specified size range for all stony and carbonaceous visited NEA. Source data are provided as a Source Data file.

as the consequence of centrifugal forces resulting from the asteroid's high spin rate. On the contrary, for Dimorphos case the spin rate is much slower[23] (11.92 h) than the one of the primary. However, the gravity is also weaker due to the body's smaller size and the tides resulting from gravitational interactions are on the same order as the centrifugal accelerations[60,61]. The tidal component of surface acceleration could then affect Dimorphos blocks' alignment also in the -E-W direction[60,61]. We, therefore, advance the idea that after the mass shedding event coming from the equatorial band of the primary and the accumulation of the secondary body, the boulders may have aligned with this preferred orientation. Nevertheless, in order to confirm such a hypothesis or not, explicit modeling of irregular-shaped particles resulting from the shedding event with gravitational torques is required[62].

Finally, when comparing the boulder number densities per km² of both Dimorphos and Didymos with those obtained from the other visited NEAs (Fig. 7A), it is possible to notice that they largely exceed all previously presented values obtained at all considered sizes (Table 1). In particular, the density per km² of Dimorphos boulders ≥1 m is 2.3x with respect to the one obtained for Bennu, while it is 3.0x with respect to Ryugu. Such values increase once Dimorphos boulders ≥5 m are compared with Bennu (3.5x), Ryugu (3.9x) and Itokawa (5.1x). At sizes ≥10 m the boulder density of Dimorphos ranges from 2.6x (Bennu) to 49.0x when compared to Eros ones. Instead, Didymos boulder densities for sizes ≥20 m range from a 5.5x when compared to Ryugu to 118.7x more than Eros. This trend is similarly reflected also for Didymos boulder densities with sizes ≥50 m. Such analysis highlights that Didymos and Dimorphos are the most boulder-rich asteroids ever visited so far. This is of particular interest in the context of asteroid studies because it could mean that contrarily to the single bodies visited so far, binary systems might be affected by subsequential fragmentation processes that largely increase their block density per km². Lastly, it is currently unknown if the Didymos boulders' SFD power-law index for sizes <22.8 m is still equal to −3.6. If so, Didymos would be globally more boulder dense at all sizes than its secondary,

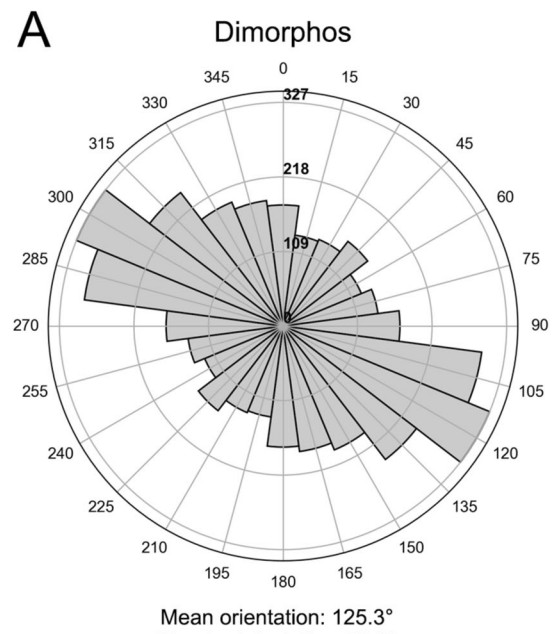

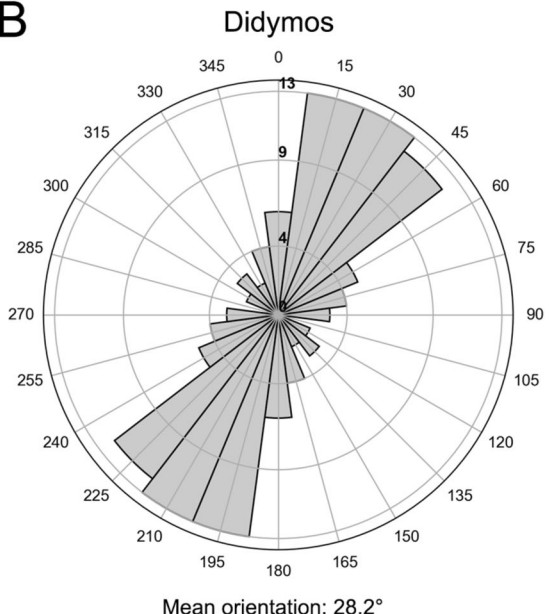

**Fig. 8 | Dimorphos and Didymos boulder orientations.** Rose diagrams of Dimorphos (**A**) boulders ≥1 m and of Didymos (**B**) boulders ≥16.5 m with an apparent axial ratio <0.9. The corresponding mean orientation and standard deviation are also indicated. We recall that Dimorphos is tidally locked with respect

to Didymos, and the hemisphere of the secondary observed by DRACO is approximately perpendicular to the Didymos facing-side[44]. Source data are provided as a Source Data file.

**Table 1 | The number density per km² of boulders ≥1 m, ≥5 m, ≥10 m, ≥20 m and ≥50 m on all visited NEAs**

| NEA | No. density/ km² ≥1 m | No. density/ km² ≥5 m | No. density/ km² ≥10 m | No. density/ km² ≥20 m | No. density/ km² ≥50 m |
|---|---|---|---|---|---|
| Dimorphos | 202,879 | 6288 | 833 | - | - |
| Didymos | - | - | - | 273 | 13.70 |
| Itokawa | - | 1234 | 158 | 15 | 1.80 |
| Eros | - | - | 17 | 2.3 | 0.11 |
| Toutatis | - | - | - | 21 | 0.40 |
| Bennu | 88,360 | 1820 | 326 | 31 | 3.78 |
| Ryugu | 68,158 | 1633 | 269 | 50 | 2.45 |

which is another evidence for the mass shedding segregation process that generated the natural satellite from the primary.

### Evidence for regolith-formation processes

The boulders that asteroids inherit from the catastrophic disruption of their parent bodies and binary formation events are expected to break up due to the effect of thermal fracturing and micrometeoroid bombardment (e.g., ref. 20,63). These mechanisms affect smaller boulders more rapidly than the large ones, hence disintegrating them and may be an alternative explanation for the Weibull SFD that we found for the boulders on Dimorphos because, as mentioned above, such a trend could be the result of different multi-fragmentation events.

Dimorphos boulders are repeatedly subject to thermal stresses which result in the formation of fractures leading to boulder failure[44]. In the suggested lifetime of Dimorphos (spanning from 0.03 to 13.3 million years[30]), this process primarily affects the smaller boulders, which are then prone to be broken apart, rather than the largest sizes that still retain the original SFD. This is due to the fact that fracture propagation needs time to occur and the larger the boulder, the longer the time to be disaggregated into smaller constituents will be needed, as shown by ref. 44. For this reason, the current Dimorphos boulder distribution (Fig. 1C) could be represented by an impact-related inherited power-law curve that was later modified and made shallower at the smaller sizes due to disintegration by thermal fracturing. In addition, it appears that on Dimorphos the median boulder size increases as the heliocentric average insolation decreases (Fig. 2F). This might suggest that bigger boulders are more likely to remain intact in regions with lower average sunlight exposure, thus experiencing less thermal alteration and consequent less formation of fractures, which could break them down.

However, even when studying boulders affected by larger heliocentric average insolation we do not observe any fines that embed or surround them[31]. On one side, this lack of fines could be simply explained by the fact that their sizes are smaller than the final full DRACO image resolution (<5.5 cm). A second explanation is linked to Dimorphos' young age, as its boulders are presently impacted by fractures that have not completely broken them apart[44], thereby not producing any fine material yet. Another possibility might involve the composition of Dimorphos, where fractured material may not necessarily result in fine deposits. However, this explanation is challenging to accept, considering that, as referenced in ref. 20 and observed on Eros[9] and Itokawa[10] (both stony NEAs like Dimorphos), such objects typically exhibit localized fine deposits.

Micrometeoroid bombardment can also be invoked to explain the shallower SFD towards the smallest sizes. Indeed, as for thermal fracturing, Dimorphos boulders are repeatedly subject to impacts. Nevertheless, recent modeling[44] highlights that impactors smaller than ~6 cm diameter occur once over 100 kyr if the impact speed is ~15 km/s (assuming that the Didymos system has stayed in the near-Earth region over this time scale[30]). If such craters produced by these small projectiles form in the strength regime, their resulting size should be of the order of a few meters[64,65]. Instead, if the impact occurs in the armouring regime[66], the size would be much smaller. Besides a couple of identified craters on Dimorphos boulders[30], the general lack of fines may suggest that an important depletion of m-size boulders due to micrometeoroid bombardment has not yet occurred on the asteroid's young surface. A second explanation for the lack of fines is that the impact ejecta velocity on stony asteroids is generally higher than carbonaceous asteroids (see ref. 67) which, combined with the small escape velocity of Dimorphos (0.09 m s⁻¹, ref. 68), may promote the loss of most of the fines produced by impacts into space.

The first-ever boulder SFD analysis of a binary NEA, as well as the blocks' number densities and orientation distributions obtained through the DART images, have returned pivotal hints on the formation and evolutionary history of the overall binary system pointing to a Didymos origin for the Dimorphos boulders. Since Didymos is part of the largest group of binary NEAs with a secondary-to-primary size ratio ranging from 0.1 to 0.6, the presented results give insights into the formation of such secondaries as a consequence of boulder shedding from the primary asteroid. Moreover, this contextualization of binary NEAs within the broader framework of small bodies enhances our understanding of their formation mechanisms and contributes to a more comprehensive perspective on the dynamics of such systems. A conclusive answer to the advanced interpretations will be provided by the upcoming Hera mission[69], which will acquire images of the full Dimorphos and Didymos bodies with spatial scales similar to the highest ones obtained by DRACO (from few to tens of cm), but from multiple observing angles and illumination geometries. Additional constraints on how Dimorphos and Didymos boulders break may come from estimation of their mechanical properties based on thermal emission measurements by the thermal imager (TIRI[69]) instrument, as previously done for carbonaceous asteroid Bennu and Ryugu (e.g., refs. 20,70,71).

## Methods

### Boulder identification, mapping, and size frequency distribution fitting

By exploiting the Johns Hopkins University Applied Physics Laboratory (JHUAPL) Small Body Mapping Tool software (SBMT[41]), we identified and mapped all visible boulders located on the surfaces of Dimorphos and Didymos. As in refs. 6,7,10,14,39,40,48, we defined as boulder a positive relief, i.e., which seems to protrude from the ground where it stands, detectable in different, increasingly higher spatial scale images with the constant presence of an elongated shadow. We highlight that following the official USGS size terms after ref. 72, "boulders" have diameters >0.25 m, "cobbles" range between 0.25 and 0.064 m, while "pebbles" sizes range between 0.064 and 0.002 m. Since in this work we considered all particles with diameters ≥1.0 m for Dimorphos, and with size ≥22.8 m for Didymos, we decided to indifferently call them boulders or blocks. In particular, through SBMT we manually fitted the boulders as ellipses, uniquely identified through the selection of three points located at the outer edge of each block. These chosen points representing the boulder limit, were easily identified on the illuminated side of the feature thanks to the well-contrasted DRACO images, showing blocks with sharp boundaries. Three closeup images (Supplementary Figs. 1–3) showing the uninterpreted surface and the boulders identified with pink ellipses have been added into the Supplementary Information. Since we are identifying all boulders larger than three pixels, the associated size-identification error does not exceed one pixel, as shown in refs. 14,39, with the corresponding SFD falling within the uncertainty[14]. Nevertheless, as mentioned in the main text, in order to be conservative on the counts and prevent potential size misinterpretation, we opted to raise the minimum size threshold deemed reliable from three to five pixels. We decided to identify all boulders as ellipses, instead of lines[6,14] or circles[10,39], because such

representation better captures the possible boulders' irregular characteristics, hence returning their maximum and minimum 2D extension[12,13]. Moreover, the identification of both the semi-major and semi-minor axis of each ellipse is pivotal in deriving the boulder apparent axial ratio, hence leading to a thorough comparison, when available, with previously visited NEAs. Afterwards, the ellipse's major axis was used as the maximum diameter of the corresponding boulder, assuming that each boulder's maximum extents are exposed and visible on the surface. The Didymos Reconnaissance and Asteroid Camera for OpNav (DRACO[38]) images we decided to use for the identification of Dimorphos boulders are those taken at distances of 52.56 km and 40.73 km, which are characterized by the highest spatial scale (0.26 and 0.20 m/pixel) covering all the visible and lit terrain of the target. Such resolutions make all boulders larger than 0.6–0.8 m identifiable. Following the same rationale, for Didymos case we used four images that were taken at distances of 990–633 km from the object with spatial scales ranging from 4.9 to 3.3 m/pixel. These resolutions provide the possibility to locate and identify all boulders larger than 10–15 m on the asteroid.

For each boulder we derived its apparent axial ratio, which is the ratio between the semi-minor axis and the semi-major axis of the ellipse. Besides the latitude and longitude, SBMT stores for the center of each boulder the gravitational slope (°), the gravitational acceleration (m/s$^2$) and the gravitational potential (J/kg), taking into account both Dimorphos's rotation as well as Didymos' tides. In addition, SBMT also derives the orientation (°) of the boulder, which is the angle between the ellipse major axis and the longitude line it lies on as projected onto the surface. To obtain the cumulative boulder size-frequency distribution per km$^2$, we made use of the corresponding areas computed from the shape model of both Dimorphos and Didymos. Afterwards, the cumulative number of boulders per km$^2$ versus size in meters are represented in a log-log plot.

The ref. 43 methodology validates the existence of the power-law fitting model which is characterized by the scaling parameter called $\alpha$. This method also allows the identification of the completeness limit $x_{\min}$, which is the threshold value above which the power-law exists. The estimation of $x_{\min}$ is done through the Kolmogorov-Smirnoff (KS) statistic and allows to find the value minimizing it. Afterwards, the parameter $\alpha$ is determined through the maximum likelihood estimator. The uncertainty for both $\alpha$ and $x_{\min}$ is then derived through a non-parametric bootstrap procedure that generates a large number of synthetic datasets from a power-law random generator and performs a number of KS tests to verify if the generated and observed data come from the same distribution. This technique returns a p-value that can be used to quantify the plausibility of the hypothesis. Given the significance level of 0.10 to validate the existence of the power-law fitting model to the data[43], if the p-value exceeds 0.10, it suggests that any deviation between the observed data and the model may be attributable to statistical variations[43]. Conversely, if the p-value falls below 0.10, this indicates that the dataset does not adhere to a power-law distribution, but rather to an alternative one[43].

In addition to the power-law fitting curve, we highlight in the main text that Dimorphos boulder sizes between 1 and 5 m show a clear departure from the power-law fit, which is not due to a resolution effect. Moreover, if we consider the full 1–16 m size range we identify that the best fitting curve to the data is the Weibull function. This function, which is in the form of $y = A \, exp^{-\left(\frac{x}{\lambda}\right)^k}$, i.e., the upper cumulative distribution function of the Weibull distribution, it is characterized by the so-called shape parameter $k > 0$ (which affects the shape of the distribution) and by the scale parameter $\lambda > 0$ (which stretches or shrinks the distribution). In the specific case where $k = 1$, the Weibull distribution becomes the exponential one. This function is largely used in fracture and fragmentation theory[46,51–53], well describing the particle distribution that is derived from grinding experiments[54].

## Data availability

The DART mission archive at NASA's Planetary Data System contains data from DRACO, as well as associated documentation and advanced products, including the shape models of Didymos and Dimorphos (https://pds-smallbodies.astro.umd.edu/data_sb/missions/dart/index.shtml and https://naif.jpl.nasa.gov/pub/naif/pds/pds4/dart/dart_spice/). The Small Body Mapping Tool (SBMT) developed by Johns Hopkins Applied Physics Laboratory contains the shape models of both asteroids with DRACO images and associated back-planes that resolve the surfaces of the asteroids (it is freely available at https://sbmt.jhuapl.edu/). The Dimorphos and Didymos boulder data generated in this study are provided in the Source Data file. Source data are provided with this paper.

## Code availability

The SBMT software developed by Johns Hopkins Applied Physics Laboratory used to identify all boulders properties indicated in the manuscript is freely available at https://sbmt.jhuapl.edu/.

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

## Acknowledgements

This work was supported by the Italian Space Agency (ASI) within the LICIACube project (ASI-INAF agreement n. 2019-31-HH.0) and HERA project (ASI-INAF agreement n. 2022-8-HH.0). This work was also supported by the DART mission, NASA Contract 80MSFC20D0004. S.C. acknowledges funding from the Crosby Distinguished Postdoctoral Fellowship Program of the Department of Earth, Atmospheric and Planetary Science, Massachusetts Institute of Technology. R.N. acknowledges support from NASA/FINESST (NNH20ZDA001N). N.M. and C.R. acknowledge funding support from the European Commission's Horizon 2020 research and innovation program under grant agreement No 870377 (NEO-MAPP project) and support from the Centre National d'Etudes Spatiales (CNES), focussed on the Hera space mission. S.D.R. acknowledges support from the Swiss National Science Foundation (project number 200021\207359). L.P. contribution was supported by the Margarita Salas postdoctoral grant funded by the Spanish Ministry of Universities—NextGenerationEU and CIAPOS/2022/066 postdoctoral grant (European Social Fund). Ö.K. acknowledges funding support from the PRODEX program managed by the European Space Agency (ESA) with help of the Belgian Science Policy Office (BELSPO). P.M. acknowledges funding support from the French space agency CNES, ESA, and The University of Tokyo. JMT-R acknowledges support from the Spanish project PID2021-128062NB-I00 funded by MCIN/AEI.

## Author contributions

M.P. conceived the work, led the project, the interpretation of the results, the development of the manuscript; M.P., F.T., A.L., O.B., identified and mapped the boulders, independently comparing the results; M.P., F.T., A.L., O.B., S.Cam., C.M.E., E.D., R.T.D., G.P., M.H., R.N., E.M.E. wrote the original draft of the manuscript and discussed the interpretation of the results; N.L.C., A.Riv., A.Che. lead the DART investigation team and provided valuable comments to the original draft; V.D.C., H.A., Y.Z., L.P., R.-L.B., S.Iva., N.M., A.Ros., C.R., S.Iev., J.B.V., F.F., S.D.R., A.C.-B., L.P., P.B., G.T., Ö.K., JMT-R., J.S., T.F., E.A., J.D.P.D., P.H.A.H., J.B., S.R.S., P.A., P.M., J.R.B., A.Z. provided valuable comments that substantially improved and revised the manuscript and the figures presented; E.D. lead the LICIACube team; M.A., S.P., G.I., I.B., A.Cap., S.Cap., M.C., G.C., M.D., I.G., L.G.C., E.G., R.L.M., M.Lav., M.Lom., D.M., P.P., D.P., P.T., M.Z., G.Z. are part of the LICIACube team and reviewed the manuscript.

## Competing interests

The authors declare no competing interests.

## Additional information

M. Pajola [1]✉, F. Tusberti [1], A. Lucchetti [1], O. Barnouin [2], S. Cambioni[3], C. M. Ernst [2], E. Dotto [4], R. T. Daly [2], G. Poggiali[5,6], M. Hirabayashi [7], R. Nakano[7,8], E. Mazzotta Epifani [4], N. L. Chabot [2], V. Della Corte[9], A. Rivkin [2], H. Agrusa [10,11], Y. Zhang [12], L. Penasa [1], R.-L. Ballouz [2], S. Ivanovski[13], N. Murdoch [14], A. Rossi [15], C. Robin[14], S. Ieva [4], J. B. Vincent [16], F. Ferrari [17], S. D. Raducan [18], A. Campo-Bagatin [19], L. Parro [19,20], P. Benavidez [19], G. Tancredi [21], Ö. Karatekin[22], J. M. Trigo-Rodriguez [23], J. Sunshine [10], T. Farnham [10], E. Asphaug[24], J. D. P. Deshapriya[4], P. H. A. Hasselmann[4], J. Beccarelli[1], S. R. Schwartz[24], P. Abell[25], P. Michel [11,26], A. Cheng [2], J. R. Brucato [5], A. Zinzi [27,28], M. Amoroso[27], S. Pirrotta [27], G. Impresario [27], I. Bertini[29], A. Capannolo[14], S. Caporali[5], M. Ceresoli[17], G. Cremonese[1], M. Dall'Ora[9], I. Gai[30,31], L. Gomez Casajus [30,31], E. Gramigna [30,31], R. Lasagni Manghi [30,31], M. Lavagna[17], M. Lombardo[30,31], D. Modenini [30,31], P. Palumbo[32], D. Perna[4], P. Tortora [30,31], M. Zannoni [30,31] & G. Zanotti [17]

[1]INAF-Astronomical Observatory of Padova, Padova, Italy. [2]Johns Hopkins University Applied Physics Laboratory, Laurel, MD, USA. [3]Department of Earth, Atmospheric and Planetary Sciences, Massachussets Institute of Technology, Cambridge, MA, USA. [4]INAF-Osservatorio Astronomico di Roma, Roma, Italy.

[5]INAF-Osservatorio Astrofisico di Arcetri, Firenze, Italy. [6]LESIA-Observatorie de Paris PSL, Paris, France. [7]Georgia Institute of Technology, Atlanta, GA, USA. [8]Department of Aerospace Engineering, Auburn University, Auburn, AL, USA. [9]INAF-Osservatorio Astronomico di Capodimonte, Napoli, Italy. [10]Department of Astronomy, University of Maryland, College Park, MD, USA. [11]Université Côte d'Azur, Observatoire de la Côte d'Azur, CNRS, Laboratoire Lagrange, Nice, France. [12]Climate & Space Sciences and Engineering, University of Michigan, Hayward, MI, USA. [13]INAF-Osservatorio Astronomico di Trieste, Trieste, Italy. [14]Institut Supérieur de l'Aéronautique et de l'Espace (ISAE-SUPAERO), Université de, Toulouse, France. [15]IFAC-CNR, Sesto Fiorentino, Firenze, Italy. [16]DLR Berlin, Berlin, Germany. [17]Dipartimento di Scienze e Tecnologie Aerospaziali, Politecnico di Milano—Bovisa Campus, Milano, Italy. [18]Space Research and Planetary Sciences, Physikalisches Institut, University of Bern, Bern, Switzerland. [19]Universidad de Alicante, de Alicante, Spain. [20]University of Arizona, Tucson, AZ, USA. [21]Dpto. Astronomia, Facultad Ciencias Igua, Montevideo, Uruguay. [22]Royal Observatory of Belgium, Uccle, Belgium. [23]Institute of Space Sciences (ICE, CSIC) and Institut d'Estudis Espacials de Catalunya (IEEC), Barcelona, Spain. [24]Planetary Science Institute; University of Arizona, Tucson, AZ, USA. [25]NASA Johnson Space Center, Houston, TX, USA. [26]Department of Systems Innovation, School of Engineering, The University of Tokyo, Tokyo, Japan. [27]Agenzia Spaziale Italiana, Roma, Italy. [28]Space Science Data Center—ASI, Roma, Italy. [29]Dipartimento di Scienze & Tecnologie, Università degli Studi di Napoli "Parthenope", Napoli, Italy. [30]Dipartimento di Ingegneria Industriale, Alma Mater Studiorum—Università di Bologna, Forlì, Italy. [31]Centro Interdipartimentale di Ricerca Industriale Aerospaziale, Alma Mater Studiorum—Università di Bologna, Forlì, Italy. [32]INAF-Istituto di Astrofisica e Planetologia Spaziali, Roma, Italy. ✉e-mail: maurizio.pajola@inaf.it

