## [Peer Review File · Nature Communications]

Evidence for multi-fragmentation and mass shedding of boulders on rubble-pile binary asteroid system (65803) DidymosREVIEWER COMMENTS

Reviewer #1 (Remarks to the Author):

Comments on the manuscript "Evidence for multi-fragmentation and mass shedding of boulders on rubble-pile binary asteroids" by Pajola et al.

The authors explore the size distribution of Dimorphos boulders smaller than 5 m, suggesting a multi-phase fragmentation process during coalescence or surface evolution. The manuscript discusses (based on a comparison of the surface distribution and shapes of boulders on Didymos and Dimorphos) the hypothesis that asteroid binary systems form through the spin-up and mass shedding of some fraction of the primary asteroid. I recommend minor revisions to the paper. Please find below the list of specific comments.

Sincerely,
Reviewer

Introduction

1. I propose slightly reorganizing the introduction. Some sentences are lengthy; it will be better to break them down into more precise and concise statements for better readability.
2. It would be better if, at first, an overview of the study's objective, followed by the importance of analyzing boulder size-frequency distribution on asteroids.

Results

1. Please describe in more detail why the phase angle of $\sim 59^\circ$ was used for the images' analysis. Discuss the quality of images.
2. Please discuss the potential influence of spatial scales on the accuracy of the obtained results.
3. Please add information about any potential limitations or assumptions associated with this approach, especially considering the irregular shapes of asteroids.
4. It is important to discuss the accuracy and potential sources of error in determining boulder sizes.

Dimorphos:

1. The authors need to clarify information about the criteria for selecting the study area on Dimorphos and the reasons behind excluding areas outside this region.
2. Please add a more detailed explanation of the rationale behind increasing the three-pixel sampling rule to five pixels and setting a lower size limit of 1.0 m for boulder inclusion.
3. Please discuss the obtained α values and their significance for analysis.
4. For better the reader's understanding, please justify the use of a Weibull function for boulders in the 1-16 m size range.
5. Fig.1. (A) Please change the color of the text and line. They are practically not identified in the image.
6. Fig.2. (A) Please change the size of the text near points. They are practically unreadable in the image.

Didymos:

1. Please write more detailed the criteria for selecting the study area on Didymos and explain why this area was chosen.
2. To analogy for Dimorphos, I propose to provide more detailed insights into the obtained α and x_{min} values.
3. It is important in this subsection to give some detailed clarification on the study area selection and careful consideration of the impact of lower spatial resolution on the obtained results.
4. Fig.4. (B) Please change the color of the lines. They are practically unidentified in the image.
5. Fig.5. (B) Please change the size of the text near points. They are practically not unidentified in the image.

Discussion. Evidence for catastrophic disruption of the parent body

1. Write more details about differences in power-law indices for stony NEAs, including Dimorphos and Didymos, compared to carbonaceous asteroids. Please add some text about the significance of these indices in the context of impact-related boulder formation and the rubble pile hypothesis.
2. I propose to compare the laboratory impact experiments' findings regarding the mean axial ratio distribution obtained for boulders on Dimorphos and Didymos in the sense of consistency with collisional disruption scenarios.
3. It would be useful to expand the explanation of the mean axial ratio trend observed in smaller Dimorphos boulders and its relation to granular processes.
4. It is needed to provide a more detailed comparison of the mean axial ratio values obtained for Dimorphos and Didymos with those of other NEAs (e.g., Eros, Itokawa, Ryugu) mentioned in Table 1 (select any notable similarities or differences).
5. Add please a small discussion about interpreting the Weibull distribution observed in boulders on Dimorphos, including the potential past or ongoing processes that may have led to this distribution.

Discussion. Evidence for formation of Dimorphos via mass shedding

1. Please add a small discussion of the possible size-dependent processes, for example, particle size segregation and comparison with the observed results.
2. Discuss in more detail the significance of the boulder number densities on Dimorphos and Didymos compared to other visited NEAs (including density ranges and sizes).
3. Write more clearly about the boulder number densities of Dimorphos and Didymos in the context of asteroid studies.

Discussion. Evidence for regolith-formation processes

1. It would be helpful to explicitly state the reasoning behind the expectation that smaller boulders are more affected by thermal stresses over the suggested lifetime of Dimorphos.
2. Please add a brief interpretation of the relationship between the median boulder size and heliocentric average insolation and its implications for the preservation of larger boulders.
3. It would be important to add a brief discussion on the implications of the observed lack of fines for our understanding of Dimorphos' surface age or composition.
4. Please add some sentences about the implications of the observed orientations for the asteroid's history or formation.
5. Similar comments also apply to color analysis. It is necessary to compare values obtained using similar apertures in km. Since the authors obtained color variations, which are important for studying the dust component of distant comets, it is essential to ensure that this is not related to the influence of changing the aperture size. I propose recalculating the color for identical apertures in km. Additionally, it would be beneficial to investigate color changes with increasing cometocentric distance separately. This study allows the authors to analyze their results more accurately.

Methods. Boulder identification, mapping, and size frequency distribution fitting

1. Please add some sentences about the significance of using ellipses to represent boulders and how this approach helps take account in capturing the boulder's characteristics.
2. In this section, it will be good to add a brief explanation of what a p-value > 0.1 or < 0.1 implies in the context of power-law distribution.

Reviewer #2 (Remarks to the Author):

The authors are analysing the size frequency distribution of boulders on the surface of each component of the NEA Didymos/Dimorphos binary asteroid that was flown (and impacted for Dimorphos) by the NASA DART spacecraft. This study is not about the aftermath of the impact, but to count the boulders, measure their sizes, ratios etc. from the high resolution images from the DRACO camera before the impact. With the goal to understand the rubble pile binary asteroids formation, and the processes shaping their surfaces. The data are coming from a rather "straightforward" detection/counting and characterisation of each boulder for each component using a specific tool, to determine the best law to fit the size distribution and interpret them at the light of what was done on the few previous high resolution NEA space observations, and to study the boulders characteristics based on their specific localisation on the asteroids surface.

The study is interesting, and the effort to measure all these boulders is important, but it does not bring clear conclusions, or is sometimes confusing, one of the main issue is the lack of models for the different scenarios proposed. It is without any doubt a complex situation and several processes might be at play, but the reader is left often to hypothetical conclusions, sometimes even contradictory conclusions (see below). It appears at the end that the power-law indexes found for both asteroids are similar to those already found for other S type NEA, while they were single asteroids compare to, in this case, a binary system. There is no clear differences highlighted between the binaries component and others (or even the fact that there is no obvious difference), so the main conclusion "asteroid binary systems form through the spin up and mass shedding of some fraction of the primary asteroid." is not totally convincing. The authors might try to clarify this.

Comments

+ The title "... rubble-pile binary asteroids" does not seem appropriate as with the plural it could be seen as a generalization, while only one binary asteroid is the purpose of this work (and the conclusion is not totally convincing without modelling).

+ What is the definition of a boulder? Or block? Is there a size limit (1m)? Everything is about boulders in the paper so a definition would be welcome.

+ "The resulting boulder SFD on the system is a powerful metric..." this sentence is coming after describing the impact and is then supposedly related to the impact? But the result is not yet known. If not, the sentence should be rephrased ("resulting" might be misleading?).

+ "... the best available resolution that fully covered the visible and illuminated surfaces of each asteroid." The purpose is clear to not be biased to one region, but there are also for Dimorphos higher resolution images (the last ones before impact). They were not used. Are they not useful for this study to have a focus on the smaller boulder sizes?

+ The resolution of the images of Dimorphos and Didymos are very different, about a factor 3 to 5. It is then not really possible to compare the same sizes range for the smallest boulders. The comparison of the boulders in the similar size range could then be more highlighted? Otherwise how this difference of resolution is affecting the comparison between both bodies? The comparison of the mode, median, and mean size of the boulders on both asteroids do not have much sense with these different resolutions.

It is noted that the Hera spacecraft should allow a better comparison but the surface of the asteroids have most probably been strongly modified.

+ Fig 1A. The grid and the blue line are nearly not visible at all, as well as the coordinates. Use a larger and higher resolution image?

+ Fig 1B. The individual boulders limits in purple are not visible. A zoom would be welcome. Idem for the grid and the resolution.

+ Fig 1C is of rather poor quality

- + The size of the largest boulders of Dimorphos is not given in the paragraph about Dimorphos
- + The Weibull function: as this function is central in the paper it would be good for the non-initiated to give in Method a bit more information/context about this function.
- + Fig 4. Same remark as for Fig 1.
- + Fig 3 and 6 do not show remarkable information, and might be dropped in Suppl. material, especially because Table 1 provides about the same information.
- + The power-law indexes are given for the NEAs observed previously, what about the Weibull function for those asteroids?
- + "Laboratory impact experiments performed with different projectile velocities, target shapes, compositions and strengths have shown..." for which target sizes? In laboratories this must be about small size objects. Can this really be extrapolated to boulder sizes above 1 m (and much bigger).
- + "The fact that both the primary and the secondary power-law fits reasonably overlap within error bars (Fig. 7) suggests that Dimorphos' surface has directly inherited part of Didymos' boulders." This affirmation and conclusion do not seem obvious. Could this be expanded? How this can be quantified? Especially if there is some segregation in the boulders mass shedding which should then deplete one of the component with respect to the other and change the size distribution between the two asteroids?
This seems to be contradictory as the two distributions are declared similar? And against the main conclusion?
- + "...even if Didymos current spin is 2.26 hr only a slightly shorter spin period of 2.2596 hr is critical for the primary to initiate surface landslides and mass shedding." Long sentence not very clear, please precise.
- + "...this mechanism would preferentially eject larger boulders from the surface." That's interesting but that seems counter-intuitive. Are we talking about boulders here or smaller particles? In any case that would produce a different size distribution on both bodies?
- + The spin rate is much longer  .. is much slower . Cite the rate.

Reviewer #3 (Remarks to the Author):

General aspects

The topic of the manuscript is to provide new observational based evidence or strong indication on the formation process of an asteroid satellite: Dymophos. The topic is important and relevant, not only from scientific point of view but also helps planning asteroid impact mitigation actions with the better understanding of asteroid interiors, surface processes and dynamic evolution of these solid bodies. The methods are moderately well described, the language is very good, the illustrations are useful and the references are relevant. There is new result presented, and the audience is waiting for such publication as it presents an important outcome of the recent DART mission. However some moderate improvements are still needed to make the work publishable, thus the referee suggests moderate revise.

Several literature sources are indicated why a specific boulder distribution etc. indicates former disruption or other processes. Some further explanation (1-2 sentences) would help many readers to get the related specific information here.

Specific aspects

around 90-95 lines

it would be useful to indicate the maximal boulder size for each mentioned bodies

108

"relatively small secondaries"

not clear what these „small secondaries“ mean

110

„Paddack effect (YORP33),“

shift the „effect“ after the bracket

120

at the end of the paragraph some expectations could be mentioned also

152

"yellow dot"

there are still persons who print the papers with black and white to read, thus suggest to make the colour coded features visible in B&W prints

166

"NEA power-law"

what is the „NEA“ for?

207

"The Didymos surface"

it would sound better „the surface of Didymos“

Figure 5

I would expect a bit more discussion on the diagram on what could be learned from it

229

"previously studied NEA global boulder SFDs"

is NEA for Near Earth Asteroids? Resolve the acronym. And would be good a short sentence on the main characteristics of these SFDs.

236

„respectively 9,10,11,16.“

suggest to finish the sentence like this: „respectively 9,10,11,16 thus represent...“ and continue the argumentation

239 „indicate an impact-related origin“

and

242 „catastrophic disruption“

not straightforward for all readers why do these aspects „imply“, suggest to briefly explain

256

„during granular processes“

suggest to cite: <https://ui.adsabs.harvard.edu/abs/2014P%26SS..101...65K/abstract>

Figure 7

suggest to make such line pattern coding (beside the colour) that allows to separate these curves on black and white prints also

286

„suggests that Dimorphos' surface has directly inherited part of Didymos' boulders“

OK, but indicate why are there smaller boulders on Dymorphos

305-306

at the „clear cut-off“

some further explanation on what does it mean and indicate would help

317-319

not very clear sentence, suggest to reformulate

320

„from random locations from Didymos“

this sentence is not clear enough, do the authors mean boulder transport from Didymos to Dymorphos much after their formation?

333

„the gravitation is“

suggest to modify to „the gravity“

and change the „smaller“ to „weaker“

Figure 8

to better understand and interpret the orientations, the relation of orientations on the two bodies relatively to each other (Dymorphos is tidally locked) should be indicated

Table 2

enlarge the text size

398

„belongs to the major binary NEA group“

what does that mean? What is the major group? Or do you mean „average object make up most of binary NEAs“?

please indicate the possible errors of the boulder size measurement in the Methods section

it would be better to have somewhere a high resolution image example on “nice” boulders on Dymorphos

REVIEWER COMMENTS

Reviewer #1 (Remarks to the Author):

Comments on the manuscript “Evidence for multi-fragmentation and mass shedding of boulders on rubble-pile binary asteroids” by Pajola et al.

The authors explore the size distribution of Dimorphos boulders smaller than 5 m, suggesting a multi-phase fragmentation process during coalescence or surface evolution. The manuscript discusses (based on a comparison of the surface distribution and shapes of boulders on Didymos and Dimorphos) the hypothesis that asteroid binary systems form through the spin-up and mass shedding of some fraction of the primary asteroid. I recommend minor revisions to the paper. Please find below the list of specific comments.

Sincerely,
Reviewer

Reply: We thank the Referee #1 for the thorough revision provided and the extremely useful suggestions advanced. We hereafter reply to all points in bold, both in this document, as well as in the Main Manuscript.

Introduction

1. I propose slightly reorganizing the introduction. Some sentences are lengthy; it will be better to break them down into more precise and concise statements for better readability.

Reply: Following the Reviewer’s suggestion we have tried to break down the lengthy sentences of the Introduction, changing them into more concise statements. All changed lines are now highlighted in bold.

2. It would be better if, at first, an overview of the study's objective, followed by the importance of analyzing boulder size-frequency distribution on asteroids.

Reply: We thank the Reviewer for this suggestion, but now that the Introduction is made of a set of more concise and precise statements, we believe that the original structure would be better to capture the flow from “Presence of Binaries in the NEA Population > Boulder SFDs, meaning and exploitation > Previously observed NEA Boulder SFDs > the (65803) Didymos Binary system, open questions about its origin > how Didymos-Dimorphos boulder SFD can help in understanding its formation scenario”. If the Referee agrees, we would like therefore to keep the structure as it is.

Results

1. Please describe in more detail why the phase angle of $\sim 59^\circ$ was used for the images' analysis. Discuss the quality of images.

Reply: The phase angle of all DRACO images has been constant throughout the full final part of the mission before impact and it was not possible to use images with a different value. Nevertheless, as showed in multiple works, as ref^{39,40,14,13}, images with phase angles in the range $40-80^\circ$ are particularly good to identify surface features as boulders and craters because they all show a constant presence of an elongated shadow (larger phase angles resulting in longer surface shadows), always in the same direction. Moreover, the DRACO images are useful for the boulders identification because they

show a well contrasted surface with features characterized by crisp boundaries. We have added 3 new lines in the main text in order to better discuss such points, that now state: “Such phase angle falls in the 40-80° range which is particularly good to identify protruding surface features as boulders^{39,40,14,13}, because they all show the presence of a well-defined shadow, which ease their identification. To perform the SFD analysis, we used the DRACO images with the best available resolution that fully covered the visible, illuminated and well-contrasted surface of each asteroid, with the presence of features characterized by crisp boundaries.”

2. Please discuss the potential influence of spatial scales on the accuracy of the obtained results.

Reply: Thanks for the suggestion. This is an important point that has been raised also by Reviewer #2. As mentioned in the new lines added to the text “We highlight that the resolution of the images on both bodies is largely different. Nevertheless, what is important for the origin and degradation implications of the two bodies is the boulder SFD trends, even at different size-ranges, and the resulting fitting curve indices^{8,10,11,13,14}”. In addition, inside the main text, we mention that we only consider trustworthy all boulders larger than five pixels, instead of the commonly used limit of three pixels^{12,39,40}. This was done in order to avoid any possible boulder misinterpretation. Nevertheless, we highlight that this is independent from the different spatial scales at play (we use a five-pixel limit for both Dimorphos, imaged at 0.20-0.26 m scale and Didymos, imaged at 3.3-4.9 m scale), as long as there are boulders large enough to be identified on the surface of the body. This is now largely explained in the new Methods section.

3. Please add information about any potential limitations or assumptions associated with this approach, especially considering the irregular shapes of asteroids.

Reply: We are not sure if the Referee is here referring to the elliptical approximation of the boulders? We think so, since after the spatial scale discussion we are mentioning the boulder ellipse identification. If this is correct, in the new Methods section we explain why we have decided to identify boulders as ellipses, instead of circles or lines, hence better capturing the possible boulders’ irregular characteristics on top of a non-flat asteroid surface. The new text in the Methods section now states “We decided to identify all boulders as ellipses, instead of lines^{6,14} or circles^{10,39}, because such representation better captures the possible boulders’ irregular characteristics, hence returning their maximum and minimum 2D extension^{12,13}. Moreover, the identification of both the semi-major and semi-minor axis of each ellipse is pivotal in deriving the boulder apparent axial ratio, hence leading to a thorough comparison, when available, with previously visited NEAs”.

4. It is important to discuss the accuracy and potential sources of error in determining boulder sizes. **Reply:** Inside the Methods section we now describe the identification of the boulders as ellipses by selecting three points located at the outer edge of each block and the error associated to it. The new text now states “In particular, through SBMT we manually fitted the boulders as ellipses, uniquely identified through the selection of three points located at the outer edge of each block. These chosen points representing the boulder limit, were easily identified on the illuminated side of the feature thanks to the well-contrasted DRACO images, showing blocks with sharp boundaries. Since we are identifying all boulders larger than three pixels, the associated size-identification error does not exceed one pixel, as showed in ref^{14,39}, with the corresponding SFD falling

within the uncertainty¹⁴. Nevertheless, as mentioned in the main text, in order to be conservative on the counts and prevent potential size misinterpretation, we opted to raise the minimum size threshold deemed reliable from three to five pixels.”

Dimorphos:

1. The authors need to clarify information about the criteria for selecting the study area on Dimorphos and the reasons behind excluding areas outside this region.

Reply: We decided to count all boulders only located inside the study area, because outside of it, all features appear distorted and stretched due to high emission angles. This means that the boulder sizes cannot be considered correct. We are here attaching four different SBMT views highlighting the point we are referring to. As it is possible to notice, outside the selected area all features appear distorted. If the Referee agrees, we would like to keep such images only in the Reply to the Reviewer document and not insert them in the Supplementary Information.

The study area considered, highlighted with the blue polygon.

As it is possible to notice, all features located outside the study area appear distorted and stretched.

As it is possible to notice, all features located outside the study area appear distorted and stretched.

2. Please add a more detailed explanation of the rationale behind increasing the three-pixel sampling rule to five pixels and setting a lower size limit of 1.0 m for boulder inclusion.

Reply: As mentioned above in Results point no. 4 “in order to be conservative on the counts and prevent potential size misinterpretation, we opted to raise the minimum size threshold deemed reliable from three to five pixels”. The new line in the main text now states “To be conservative on the boulder counts, and avoid any possible boulder size misinterpretation which often happens at the smallest dimensions¹⁴, we decided to increase the three-pixel sampling rule to five-pixels, i.e. setting a lower size limit of a boulder that is 1.0 m in size to our data”. For the sake of completeness, such line has also been changed in the Didymos section.

3. Please discuss the obtained α values and their significance for analysis.

Reply: The Dimorphos section is just intended to present the results obtained, as the power-law index α of -3.4 ± 1.3 and x_{\min} (a description of what α and x_{\min} are, in the context of the ref⁴³ methodology is presented in the Methods section). A full and detailed discussion of this value and its significance on the formative process of the studied body is presented in the *Discussion- Evidence for catastrophic disruption of the parent body* that we modified as follows “The Dimorphos power-law index α of -3.4 ± 1.3 , obtained for boulders ≥ 5 m, and Didymos power-law index α of -3.6 ± 0.7 , derived from boulder sizes ≥ 22.8 m, confirm this generally steeper stony boulder SFD (Fig. 7B) when compared to the carbonaceous one, as well as they indicate a sudden, impact-related origin for the identified boulders^{10,11,16}. As for the other visited bodies, this evidence, coupled with the maximum identified boulder dimensions (93 m on Didymos and 16 m on Dimorphos) that both exceed 1/10 the NEAs’ diameters, imply that such asteroids are collections of debris resulting from the catastrophic breakup of a larger parent body^{16,8,30,31}, followed by the reaccretion of part of its fragments”. In order to avoid possible repetitions and to keep the Results as synthetic as possible we would like to keep the text with this structure, if the Referee agrees.

4. For better the reader's understanding, please justify the use of a Weibull function for boulders in the 1-16 m size range.

Reply: As for the previous point, in the Results section of Dimorphos we only show that the best fitting curve to the SFD data is the Weibull function. Nevertheless, why this is used, as well as its formative and degradation implication for Dimorphos is largely described in the Discussion part of the manuscript *Evidence for catastrophic disruption of the parent body* as well as *Evidence for regolith-formation processes*. In addition, the full description of the Weibull function is presented in the last eight lines of the Methods section that state “Moreover, if we consider the full 1-16 m size range we identify that the best fitting curve to the data is the Weibull function. This function, which is in the form of $y = A \exp\left(-\left(\frac{x}{\lambda}\right)^k\right)$, i.e. the upper cumulative distribution function of the Weibull distribution, it is characterized by the so-called shape parameter $k > 0$ (which affects the shape of the distribution) and by the scale parameter $\lambda > 0$ (which stretches or shrinks the distribution). In the specific case where $k = 1$, the Weibull distribution becomes the exponential one. This function is largely used in fracture and fragmentation theory^{52,51,46,53}, well describing the particle distribution that is derived from grinding experiments⁵⁴”.

5. Fig.1. (A) Please change the color of the text and line. They are practically not identified in the image.

Reply: Following the Reviewer’s suggestion we have prepared a new Fig. 1 better highlighting both the study area limit (in blue), the ellipses representing the boulders (in pink), and the text. In order to be visible in b/w images, we have indicated the DART

impact site with a white triangle. All panels are now on top of each other in order to show a much larger Figure.

6. Fig.2. (A) Please change the size of the text near points. They are practically unreadable in the image.

Reply: Following the Reviewer's suggestion we have prepared a new figure with a much larger text near the points.

Didymos:

1. Please write more detailed the criteria for selecting the study area on Didymos and explain why this area was chosen.

Reply: As done for Dimorphos, we decided to count all boulders only located inside the study area (highlighted in blue), because outside of it, all features appear distorted and stretched due to high emission angles. This means that the boulder sizes cannot be considered correct.

The study area considered, highlighted with the blue polygon.

As it is possible to notice, all features located outside the study area appear distorted and stretched.

As it is possible to notice, all features located outside the study area appear distorted and stretched.

2. To analogy for Dimorphos, I propose to provide more detailed insights into the obtained α and x_{\min} values.

Reply: As for the Dimorphos section also the Didymos *Results* one is intended to solely present the results obtained, as the power-law index α of -3.6 ± 0.7 and x_{\min} (a description of what α and x_{\min} are, in the context of the ref⁴³ methodology is presented in the Methods section). A full and detailed discussion of this value and its significance on the formative process of the studied body is presented in the Discussion- Evidence for catastrophic disruption of the parent body that we modified as follows “The Dimorphos power-law index α of -3.4 ± 1.3 , obtained for boulders ≥ 5 m, and Didymos power-law index α of -3.6 ± 0.7 , derived from boulder sizes ≥ 22.8 m, confirm this generally steeper stony boulder SFD (Fig. 7B) when compared to the carbonaceous one, as well as they indicate a sudden, impact-related origin for the identified boulders^{10,11,16}. As for the other visited bodies, this evidence, coupled with the maximum identified boulder dimensions (93 m on Didymos and 16 m on Dimorphos) that both exceed 1/10 the NEAs’ diameters, imply that such asteroids are collections of debris resulting from the catastrophic breakup of a larger parent body^{16,8,30,31}, followed by the reaccretion of part of its fragments”. In order to avoid possible repetitions and to keep the Results as synthetic as possible we would like to keep the text with this structure, if the Referee agrees.

3. It is important in this subsection to give some detailed clarification on the study area selection and careful consideration of the impact of lower spatial resolution on the obtained results.

Reply: We have decided to consider only this area of Didymos because outside of it, all features appear distorted and stretched due to high emission angles. This is highlighted right at the beginning of the Didymos subsection together with the images above showing the boulder distortion effect.

Regarding the impact of lower spatial scale images, it is clear that this results in a smaller number of boulders identified, with respect to Dimorphos. Nevertheless, a block SFD with a significative power-law fit is derived, as showed by the results obtained using the ref⁴³ methodology and the p-value of 0.6 associated.

This is now highlighted better in the text that states: “From Fig. 4C the cumulative number of boulders per km^2 is well fit by a power-law curve with $\alpha = -3.6 \pm 0.7$ and $x_{\min} = 22.8 \pm 2.3$ m. The p-value derived from 2500 Kolmogorov-Smirnoff statistical tests is 0.6, i.e. well above the 0.1 significance level. Such result shows that despite a lower spatial scale of the DRACO imagery dataset for Didymos, a boulder SFD with a significative fit and associated index is derived.”

4. Fig.4. (B) Please change the color of the lines. They are practically unidentified in the image.

Reply: Following the Reviewer’s suggestion we have prepared a new Fig. 4 better highlighting the study area limit (in blue), the ellipses representing the boulders (in pink), and the text. As for Fig. 1, all panels are now on top of each other in order to show a much larger Figure.

5. Fig.5. (B) Please change the size of the text near points. They are practically not unidentified in the image.

Reply: Following the Reviewer’s suggestion we have prepared a new figure with a much larger text near the points.

Discussion. Evidence for catastrophic disruption of the parent body

1. Write more details about differences in power-law indices for stony NEAs, including Dimorphos and Didymos, compared to carbonaceous asteroids. Please add some text about the significance of these indices in the context of impact-related boulder formation and the rubble pile hypothesis.

Reply: Thanks for the suggestion. We have now added more details to the text that states: “As shown in Fig. 7B, boulders on all stony NEAs previously visited by spacecrafts—Itokawa, Eros and Toutatis—are characterized by power-law fitting curves with indices steeper than -3.0. In particular, Itokawa shows a power-law index of -3.05 ± 0.14 for boulders $\geq 5 \text{ m}^{10}$, Eros has a power-law index of -3.25 ± 0.14 for boulders $\geq 10 \text{ m}^9$, while Toutatis is characterized by a power-law index of -4.4 ± 0.1 for boulders $\geq 20 \text{ m}^{11}$. On the contrary, on carbonaceous asteroids Ryugu and Bennu the power-law indexes obtained are -2.65 ± 0.05 for boulders $\geq 5 \text{ m}^{13}$ and -2.5 ± 0.1 for boulders with sizes $\geq 0.2 \text{ m}^{14}$, respectively. Such indices all confirm an impact-related formation that led to an SFD characterized by fractals⁴⁶. However, the variance in trends between them confirms that the power-law index is greater among stony asteroids compared to carbonaceous ones. This underscores the distinct responses of materials (stony versus carbonaceous) to meteoroid impacts and thermal cracking, as previously suggested by ref²⁰. The Dimorphos power-law index α of -3.4 ± 1.3 , obtained for boulders $\geq 5 \text{ m}$, and Didymos power-law index α of -3.6 ± 0.7 , derived from boulder sizes $\geq 22.8 \text{ m}$, confirm this generally steeper stony boulder SFD (Fig. 7B) when compared to the carbonaceous one, as well as they indicate a sudden, impact-related origin for the identified boulders^{10,11,16}. As for the other visited bodies, this evidence, coupled with the maximum identified boulder dimensions (93 m on Didymos and 16 m on Dimorphos) that both exceed 1/10 the NEAs’ diameters, imply that such asteroids are collections of debris resulting from the catastrophic breakup of a larger parent body^{16,8,30,31}, followed by the reaccretion of part of its fragments”.

2. I propose to compare the laboratory impact experiments' findings regarding the mean axial ratio distribution obtained for boulders on Dimorphos and Didymos in the sense of consistency with collisional disruption scenarios.

Reply: The full section related to laboratory impact experiments' findings and the apparent mean axial ratio distribution on Dimorphos and Didymos is intended to show the consistency with the collisional disruption scenarios. We highlight this at the end of the analysis discussion where we state: “The mean axial ratio values found from the analysis are in agreement with laboratory experiments and consistent with the other NEAs, hence indicating that both Didymos and Dimorphos blocks are the result of the catastrophic disruption of their parent body”.

3. It would be useful to expand the explanation of the mean axial ratio trend observed in smaller Dimorphos boulders and its relation to granular processes.

Reply: Following the Reviewer suggestion we have expanded the text than now states “Such behavior is explained by the fact that smaller boulders have a vertical axis that gradually becomes perpendicular to the surface during granular processes⁵⁰, owing to their lower friction angle and gravitational stability. Indeed, as mentioned in ref¹⁶, once the reaccretion process has occurred, smaller boulders are redistributed due to seismic shaking caused by repeated impacts. This is favored by their higher mobility due to the lower friction angle. Such migration consequently affects their orientation, letting them “laying flat” on the surface. This is the reason why the apparent axial ratio of smaller

boulders tends to approach the one of laboratory fragments as the size of small boulders decreases¹⁶”.

4. It is needed to provide a more detailed comparison of the mean axial ratio values obtained for Dimorphos and Didymos with those of other NEAs (e.g., Eros, Itokawa, Ryugu) mentioned in Table 1 (select any notable similarities or differences).

Reply: We have positively followed the Reviewer’s suggestion. We have added new text inside the Discussion that now states: “The mean axial ratios values found from the analysis are in agreement with laboratory experiments and consistent with the other NEAs, hence indicating that both Didymos and Dimorphos blocks are the result of the catastrophic disruption of their parent body. Nevertheless, if we interpret the mean axial ratio values for boulders ≥ 5.0 m on Dimorphos, Itokawa and Ryugu, we clearly notice that on the latter, a much higher surface mobility has occurred making them appear almost parallel to the surface (mean axial ratio of 0.68). On the contrary, Itokawa boulders ≥ 5.0 m appear in an intermediate situation (0.63), while Dimorphos boulders ≥ 5.0 m seem to not have been affected by any surface movement yet, considering that the 0.53 value is the smallest obtained. On the contrary, Eros boulders ≥ 30.0 m (as for Didymos case) have an apparent axial ratio which is 0.72, hence indicating that they are parallel to the surface”.

5. Add please a small discussion about interpreting the Weibull distribution observed in boulders on Dimorphos, including the potential past or ongoing processes that may have led to this distribution.

Reply: The Weibull distribution is commonly thought to result from sequential fragmentation⁵¹, and it is widely used in fracture theory^{52,51,46,53,48}. In addition, such distribution is often used to describe the particle distribution that is derived from grinding experiments⁵⁴, i.e., a multiple-event fragmentation. As stated at the end of the subsection “Dimorphos Weibull boulder SFD could be explained by a multi-phase fragmentation process related to the boulder’s relative velocities and sizes, which occurred during the accretion of the secondary”. A thorough explanation of this is detailed in the following subsection *Evidence for formation of Dimorphos via mass shedding*.

Discussion. Evidence for formation of Dimorphos via mass shedding

1. Please add a small discussion of the possible size-dependent processes, for example, particle size segregation and comparison with the observed results.

Reply: Following the Reviewer’s suggestion we have now changed the text that states: “Moreover, the occurrence of landslides and mass shedding from the primary while forming the secondary due to rapid spin-up caused by the YORP effect^{34,55} is also one of the possible explanations of the resulting Weibull Dimorphos SFD, since they may result in specific particle size-segregation and/or size-depletion. Hence, such processes have a significant role in shaping the final surface appearance of low-gravity bodies, since they can cause the formation of localized deposits characterized by different texture, or they can result in particle SFD which are characterized by multiple curves, i.e. different formative and/or degradation events. In particular, there is evidence that particle size segregation occurs during shallow gravity-driven surface flows and that the larger particles tend to migrate to the top and attain higher speed in a freely flowing granular system, see, e.g. ref^{56,57}”.

2. Discuss in more detail the significance of the boulder number densities on Dimorphos and Didymos compared to other visited NEAs (including density ranges and sizes).

Reply: The reply to this point is coupled and detailed with the following one.

3. Write more clearly about the boulder number densities of Dimorphos and Didymos in the context of asteroid studies.

Reply to the previous point raised coupled with this one: Following the Reviewer suggestions we have re-written the section about the boulder number densities of Dimorphos and Didymos and compared to other visited NEAs. The text now states: “Finally, when comparing the boulder number densities per km² of both Dimorphos and Didymos with those obtained from the other visited NEAs (Fig. 7A), it is possible to notice that they largely exceed all previously presented values obtained at all considered sizes (Table 2). In particular, the density per km² of Dimorphos boulders ≥ 1 m is 2.3x with respect to the one obtained for Bennu, while it is 3.0x with respect to Ryugu. Such values increase once Dimorphos boulders ≥ 5 m are compared with Bennu (3.5x), Ryugu (3.9x) and Itokawa (5.1x). At sizes ≥ 10 m the boulder density of Dimorphos range from 2.6x (Bennu) to 49.0x when compared to Eros ones. Instead, Didymos boulder densities for sizes ≥ 20 m range from a 5.5x when compared to Ryugu to 118.7x more than Eros. This trend is similarly reflected also for Didymos boulder densities with sizes ≥ 50 m. Such analysis highlights that Didymos and Dimorphos are the most boulder-rich asteroids ever visited so far. This is of particular interest in the context of asteroid studies because it could mean that contrarily to the single bodies visited so far, binary systems are affected by subsequential fragmentation processes that largely increase their block density per km². Lastly, it is currently unknown if the Didymos boulders’ SFD power-law index for sizes < 22.8 m is still equal to -3.6. If so, Didymos would be globally more boulder dense at all sizes than its secondary, which is another evidence for the mass shedding segregation process that generated the natural satellite from the primary”.

Discussion. Evidence for regolith-formation processes

1. It would be helpful to explicitly state the reasoning behind the expectation that smaller boulders are more affected by thermal stresses over the suggested lifetime of Dimorphos.

Reply: Small and large boulders are all affected by thermal stresses. Nevertheless, as shown by ref⁴⁴, in an equal timeframe, smaller blocks will be more easily disaggregated into smaller components rather than the big boulders. This is due to the fact that fracture propagation needs time to occur and the larger the boulder the longer the time to be disaggregated will be needed⁴⁴. We thank the Reviewer for giving us the opportunity to better express ourselves. We have now changed the main text that states: “In the suggested lifetime of Dimorphos (spanning from 0.03 to 13.3 million years³⁰), this process primarily affects the smaller boulders, which are then prone to be broken apart, rather than the largest sizes that still retain the original SFD. This is due to the fact that fracture propagation needs time to occur and the larger the boulder, the longer the time to be disaggregated into smaller constituents will be needed, as shown by ref⁴⁴. For this reason,…”

2. Please add a brief interpretation of the relationship between the median boulder size and heliocentric average insolation and its implications for the preservation of larger boulders.

Reply: As mentioned in the main text, the larger boulders detected on Dimorphos are located on areas that receive less heliocentric average insolation, hence resulting in a smaller thermal alteration. This can be then interpreted as an effect of smaller thermal fracturing occurring on such blocks, which can more easily pertain their original sizes. We have now changed the text that states: “In addition, it appears that on Dimorphos the median boulder size increases as the heliocentric average insolation decreases (Fig. 2F). **This might suggest that bigger boulders are more likely to remain intact in regions with lower average sunlight exposure, thus experiencing less thermal alteration and consequent less formation of fractures, which could break them down**”.

3. It would be important to add a brief discussion on the implications of the observed lack of fines for our understanding of Dimorphos' surface age or composition.

Reply: Following the Reviewer's suggestion we have changed the text that now states: “**On one side, this lack of fines could be simply explained by the fact that their sizes are smaller than the final full DRACO image resolution (< 5.5 cm). A second explanation is linked to Dimorphos' young age, as its boulders are presently impacted by fractures that have not completely broken them apart⁴⁴, thereby not producing any fine material yet. Another possibility might involve the composition of Dimorphos, where fractured material may not necessarily result in fine deposits. However, this explanation is challenging to accept, considering that, as referenced in ref²⁰ and observed on Eros⁹ and Itokawa¹⁰ (both stony NEAs like Dimorphos), such objects typically exhibit localized fine deposits**”.

4. Please add some sentences about the implications of the observed orientations for the asteroid's history or formation.

Reply: The observed boulder orientations are discussed in the *Evidence for formation of Dimorphos via mass shedding* section, where we discussed about the possible implications that the observed boulder orientations have about the asteroids' formation and evolution. We hereafter present what the text states: “**One possible explanation for Didymos blocks' alignment could be a preferential NNE-SSW regolith migration, already noted on Bennu⁵⁹ as the consequence of centrifugal forces resulting from the asteroid's high spin rate. On the contrary, for Dimorphos case the spin rate is much slower²³ (11.92 h) than the one of the primary. However, the gravity is also weaker due to the body's smaller size and the tides resulting from gravitational interactions are on the same order as the centrifugal accelerations^{60,61}. The tidal component of surface acceleration could then affect Dimorphos blocks' alignment also in the ~E-W direction^{60,61}. We therefore advance the idea that after the mass shedding event coming from the equatorial band of the primary and the accumulation of the secondary body, the boulders may have aligned with this preferred orientation. Nevertheless, in order to confirm such a hypothesis or not, explicit modelling of irregular-shaped particles resulting from the shedding event with gravitational torques is required⁶².**”

5. Similar comments also apply to color analysis. It is necessary to compare values obtained using similar apertures in km. Since the authors obtained color variations, which are important for studying the dust component of distant comets, it is essential to ensure that this is not related to the influence of changing the aperture size. I propose recalculating the color for identical apertures in km. Additionally, it would be beneficial to investigate color changes

with increasing cometocentric distance separately. This study allows the authors to analyze their results more accurately.

Reply: We thank the Reviewer for this comment, but we have not focused on a color analysis in this work, nor discussed the dust component of distant comets using aperture photometry. We believe that the sentences of point 5 are maybe related to another paper/topic?

Methods. Boulder identification, mapping, and size frequency distribution fitting

1. Please add some sentences about the significance of using ellipses to represent boulders and how this approach helps take account in capturing the boulder's characteristics.

Reply: We have added 6 new lines in order to explain why we decided to identify boulders as ellipses, instead of lines or circles. The text now states: “In particular, through SBMT we manually fitted the boulders as ellipses, uniquely identified through the selection of three points located at the outer edge of each block. These chosen points representing the boulder limit, were easily identified on the illuminated side of the feature thanks to the well-contrasted DRACO images, showing blocks with sharp boundaries. Since we are identifying all boulders larger than three pixels, the associated size-identification error does not exceed one pixel, as showed in ref^{14,39}, with the corresponding SFD falling within the uncertainty¹⁴. Nevertheless, as mentioned in the main text, in order to be conservative on the counts and prevent potential size misinterpretation, we opted to raise the minimum size threshold deemed reliable from three to five pixels. We decided to identify all boulders as ellipses, instead of lines^{6,14} or circles^{10,39}, because such representation better captures the possible boulders’ irregular characteristics, hence returning their maximum and minimum 2D extension^{12,13}. Moreover, the identification of both the semi-major and semi-minor axis of each ellipse is pivotal in deriving the boulder apparent axial ratio, hence leading to a thorough comparison, when available, with previously visited NEAs.”

2. In this section, it will be good to add a brief explanation of what a p-value > 0.1 or < 0.1 implies in the context of power-law distribution.

Reply: We have added a brief explanation in the Methods section changing the previous sentence. This is done to better explain what the 0.1 p-value means in order to validate the existence of the power-law fitting model to the data. The text now states: “The uncertainty for both α and x_{\min} is then derived through a non-parametric bootstrap procedure that generates a large number of synthetic datasets from a power-law random generator and performs a number of KS tests to verify if the generated and observed data come from the same distribution. This technique returns a p-value that can be used to quantify the plausibility of the hypothesis. Given the significance level of 0.10 to validate the existence of the power-law fitting model to the data⁴³, if the p-value exceeds 0.10, it suggests that any deviation between the observed data and the model may be attributable to statistical variations⁴³. Conversely, if the p-value falls below 0.10, this indicates that the dataset does not adhere to a power-law distribution, but rather to an alternative one⁴³”.

Reviewer #2 (Remarks to the Author):

The authors are analysing the size frequency distribution of boulders on the surface of each component of the NEA Didymos/Dimorphos binary asteroid that was flown (and impacted for Dimorphos) by the NASA DART spacecraft. This study is not about the aftermath of the impact, but to count the boulders, measure their sizes, ratios etc. from the high-resolution images from the DRACO camera before the impact. With the goal to understand the rubble pile binary asteroids formation, and the processes shaping their surfaces. The data are coming from a rather "straightforward" detection/counting and characterisation of each boulder for each component using a specific tool, to determine the best law to fit the size distribution and interpret them at the light of what was done on the few previous high resolution NEA space observations, and to study the boulders characteristics based on their specific localisation on the asteroids surface.

The study is interesting, and the effort to measure all these boulders is important, but it does not bring clear conclusions, or is sometimes confusing, one of the main issue is the lack of models for the different scenarios proposed. It is without any doubt a complex situation and several processes might be at play, but the reader is left often to hypothetical conclusions, sometimes even contradictory conclusions (see below). It appears at the end that the power-law indexes found for both asteroids are similar to those already found for other S type NEA, while they were single asteroids compare to, in this case, a binary system. There is no clear differences highlighted between the binaries component and others (or even the fact that there is no obvious difference), so the main conclusion "asteroid binary systems form through the spin up and mass shedding of some fraction of the primary asteroid." is not totally convincing.

The authors might try to clarify this.

Reply: We thank the Reviewer #2 for the important concerns raised and for the comments, suggestions and corrections received. Thanks to this revision and to the comments received by the other two Reviewers, we hope to have improved the manuscript aiming to both clarify our findings and to avoid non-clear conclusions. We have added further details inside each Discussion subsection to better explain our results.

Firstly, we have modified the Introduction to better introduce the topic and state what we expect on a binary system considering the formation scenarios proposed. In this way, we pose the open questions at the beginning of the manuscript, hence clarifying that the aim of this analysis is to quantitatively test such hypotheses. We have added the following lines at the end of the Introduction section: "... One formation hypothesis of such binary bodies is that due to the Yarkovsky–O’Keefe–Radzievskii–Paddack (YORP³³) effect, a larger primary might have experienced continuous spin-up to reach its spin limit. As a consequence, a mass shedding event or fission of some fraction of its body^{26,34,35} occurred. Ejected materials from the primary are predicted to remain in orbit within the system and reaccumulate outside the Roche limit into a small satellite. If the formation of Dimorphos is related to the top shape and rapid spin-up of Didymos by the YORP effect^{26,34,36,37}, it is expected that its boulders previously belonged to the equatorial region of the primary and have a comparable, inherited SFD. Moreover, if this interpretation holds true, we could also expect some sort of equatorial block

depletion on Didymos, as a result of the spin-up process, followed by the mass-shedding event. Here we quantitatively test this ...”.

Then, as stated by the Reviewer, our findings show that the power-law indexes found for both asteroids are similar to those already found for other S type NEA, while they were single asteroids compare to, in this case, a binary system.

Considering the complex dynamic that occurred within the binary system, we highlight that this is already an important result. Indeed, before having the first images of a binary system ever, it was not clear what to expect, and the fact that we have observed such a trend has important formative implications. In addition, when comparing the boulder number densities per km² of both Dimorphos and Didymos with those obtained from the other visited NEAs (Fig. 7A), it is possible to notice that they largely exceed all previously presented values obtained at all considered sizes (Table 1 in the Manuscript). This is of particular interest in the context of asteroid studies because it could mean that contrarily to the single bodies visited so far, binary systems are affected by subsequential fragmentation processes that largely increase at all sizes their block density per km². We have reported and expanded the text of the Discussion section to better clarify the point (please, see the Reply to each single point raised below).

Within this analysis we have also found that both asteroids' power-law fits reasonably overlap within error bars (Fig. 7). By assuming that the primary body is characterized by a boulder SFD inherited by the catastrophic disruption of the parent body, at some point, due to the rapid spin-up caused by the YORP effect the spin of Didymos reaches the critic value of 2.25 hr, hence triggering surface landslides and mass shedding. Due to the Coriolis effect, larger boulders tend to acquire larger kinetic energy and slide faster, but we underscore that all surface boulders are affected by such mass movements. If this is the case, then the inherited boulder population of the secondary object formed should be characterized by a similar, inherited SFD from the primary, at least for the bigger sizes. This is exactly what we have observed on the primary-Didymos and on the secondary-Dimorphos. On the contrary, the number decrease we see in the 1-5 m size range best fit by a Weibull curve (on Dimorphos), could be the result of fragmentation occurring right when the mass shedding from the primary was happening. This is the first time that we can observe this effect on the boulder SFD, and it was possible only thanks to the simultaneous analysis of both bodies in the system.

We modified the text accordingly, hoping that such explanation is clearer than before, and aiming to avoid any contradictory conclusions.

Comments

+ The title "... rubble-pile binary asteroids" does not seem appropriate as with the plural it could be seen as a generalization, while only one binary asteroid is the purpose of this work (and the conclusion is not totally convincing without modelling).

Reply: We agree with the Referee. For this reason, we have changed the title that now states: “Evidence for multi-fragmentation and mass shedding of boulders on rubble-pile binary asteroid (65803) Didymos-Dimorphos”. If the Referee agrees, we would like to keep this new version rather than the previous one.

+ What is the definition of a boulder? Or block? Is there a size limit (1m)? Everything is about boulders in the paper so a definition would be welcome.

Reply: Thanks for the suggestion. At the beginning of the manuscript, after the abstract, we have now added three lines about the boulder and block definition: “We hereafter use blocks and boulders as synonyms, being positive reliefs^{6,7} larger than 0.25 m, which seem to protrude from the ground where they stand, and detectable in different, increasingly higher spatial scale images with the constant presence of an elongated shadow, see Methods.” Moreover, we have added 7 lines in the Methods section, introducing the Wentworth (1922) USGS definition for boulders, cobbles and pebbles (new reference #72). The text now states: “As in ref^{6,7,10,14,39,40,48}, we defined as boulder a positive relief, i.e., which seems to protrude from the ground where it stands, detectable in different, increasingly higher spatial scale images with the constant presence of an elongated shadow. We highlight that following the official USGS size terms after ref⁷², ‘boulders’ have diameters >0.25 m, ‘cobbles’ range between 0.25 and 0.064 m, while ‘pebbles’ sizes range between 0.064 and 0.002 m. Since in this work we considered all particles with diameters ≥ 1.0 m for Dimorphos, and with size ≥ 22.8 m for Didymos, we decided to indifferently call them boulders or blocks”.

+ "The resulting boulder SFD on the system is a powerful metric..." this sentence is coming after describing the impact and is then supposedly related to the impact? But the result is not yet known. If not, the sentence should be rephrased ("resulting" might be misleading?).

Reply: The Referee is right, the term “resulting” is misleading and the sentence is not supposed to be related to the impact, since the boulder SFD we study is of the pre-impacted surface. We have changed the sentence that now states: “The boulder SFD of the system, obtained through DART high-resolution images, is therefore a powerful metric for distinguishing among previously proposed formation scenarios of binary asteroids^{26,27,28,29}”.

+ "... the best available resolution that fully covered the visible and illuminated surfaces of each asteroid." The purpose is clear to not be biased to one region, but there are also for Dimorphos higher resolution images (the last ones before impact). They were not used. Are they not useful for this study to have a focus on the smaller boulder sizes?

Reply: This is correct. The idea we had was to study the widest possible regions of both asteroids in order i) not to focus on a smaller area that could be biased, for example, due to a specific geological setting, and ii) to make comparisons with all previous NEA global boulder SFD studies. Nevertheless, by the time we have written this manuscript, we had already counted all cobbles and boulders on the full final DRACO image before impact, extracting their maximum width through the ARCGIS software (see the following image).

Boulder-cobble identification of the DART impact site location. The image used is the last, full-frame DRACO 5.5 cm spatial scale one before the spacecraft disintegration.

The resulting SFD, obtained by merging the global Dimorphos counts, and the local/impact site counts is hereafter presented, together with the best-fit obtained.

The resulting boulder-cobble SFD obtained by merging the global and local (impact site) Dimorphos counts. The dashed blue line represents the 5-pixel limit based on the final 5.5 cm resolution DRACO image.

As shown in the figure above, the Weibull curve is still the best fit to the overall counts, while a power-law fit does not work when taking into account the smallest sizes. This result is similar to the one presented in Fig. 1C, but with a different lower-size limit. Since one of our JHUAPL colleague is currently working on a manuscript fully dedicated to the impact site (the impact site boulder analysis will be one of the main sections), we decided to not insert such figure in the manuscript and fully focus on the global counts, also considering the consistency between the results.

+ The resolution of the images of Dimorphos and Didymos are very different, about a factor 3 to 5. It is then not really possible to compare the same sizes range for the smallest boulders. The comparison of the boulders in the similar size range could then be more highlighted? Otherwise how this difference of resolution is affecting the comparison between both bodies? The comparison of the mode, median, and mean size of the boulders on both asteroids do not have much sense with these different resolutions.

It is noted that the Hera spacecraft should allow a better comparison but the surface of the asteroids have most probably been strongly modified.

Reply: This is correct. The problem with such comparison between the smallest sizes for Didymos (with sizes < 22.8 m) and the largest ones for Dimorphos (> 16 m), is that on the latter we do not have counts much larger than 16 m. We see some larger boulders at the Dimorphos limb, but the images and the existing viewing angles are so stretched that a proper size measurement would be misleading. This will be different once we will have Hera images of the full surface, but as the Referee points out, the surface of the body could be heavily modified. On the contrary, the DRACO images of Didymos are much lower spatial scale when compared to the Dimorphos ones, hence leading to a very difficult identification of the smallest sizes that could be used for a comparison with Dimorphos. Nevertheless, what is important to highlight is that despite slightly different size-ranges (≤ 16 m and ≥ 22.8 m), both SFDs are characterized by curves that show comparable power-law indices. This is extremely important because it suggests that the Dimorphos SFD might have been inherited from the primary, which was one of the main open questions regarding boulders' formation.

In order to better explain this point and to highlight the largely different resolution of the images we have added 3 new lines in in the Results section that now state “We highlight that the resolution of the images on both bodies is largely different.

Nevertheless, what is important for the origin and degradation implications of the two bodies is the boulder SFD trends, even at different size-ranges, and the resulting fitting curve indices^{8,10,11,13,14}.”, as well as four new lines in the first part of the “*Evidence for formation of Dimorphos via mass shedding*” section in order to, hopefully, better explain this point: “Although the fitting ranges for Dimorphos and Didymos are slightly different, i.e. ≤ 16 m for the secondary, while ≥ 22.8 m for the primary, the fact that both asteroids' power-law fits reasonably overlap within error bars (Fig. 7) suggests that Dimorphos' surface has directly inherited part of Didymos' boulder SFD.”.

Regarding the mode, median and mean of the boulders on both asteroids: they are indicated in order to provide quantitative properties of the two distributions we found, but they were not presented to make comparisons between the two. Instead, what is extremely important to be compared is the similar SFD obtained and the resulting density of boulders per km².

+ Fig 1A. The grid and the blue line are nearly not visible at all, as well as the coordinates. Use a larger and higher resolution image?

Reply: Following the Reviewer's suggestion we have changed Fig. 1A, B and C, better highlighting the study area with the blue polygon, the ellipses with a pink, thicker outline, as well as making the coordinates bigger. We hope that the figure is much more readable now.

+ Fig 1B. The individual boulders limits in purple are not visible. A zoom would be welcome. Idem for the grid and the resolution.

Reply: See Reply for the previous comment. Regarding a zoom, as requested by both Reviewer 2 and 3, we decided to add three closeup images in the Supplementary Information where both the uninterpreted surface, as well as the identified boulders are showed.

+ Fig 1C is of rather poor quality

Reply: We are sorry for that. In the original manuscript there were high-resolution Figures that could be zoomed in. We believe that the conversion into pdf might have

affected that. We have prepared a new Fig. 1 with all images one on top of the other. In this way they are bigger and should be of much higher quality.

+ The size of the largest boulders of Dimorphos is not given in the paragraph about Dimorphos

Reply: We are not sure we correctly understood the point raised by the Referee. Inside Dimorphos section we state: “On Dimorphos, we identified a study area of 0.0132 km², outside of which all surface features appear distorted and stretched due to high emission angles (Fig. 1A). Within this area we counted 4734 boulders (Fig. 1B), finding a maximum size of 16 m.” The size of the largest boulder is here indicated. Is there a specific part of the text where the Referee suggests this number should be added?

+ The Weibull function: as this function is central in the paper it would be good for the non-initiated to give in Method a bit more information/context about this function.

Reply: Thanks for the important suggestion. We have now added 7 new lines at the end of the Methods section providing, we hope, more information and context about this function. The text now states: “In addition to the power-law fitting curve, we highlight in the main text that Dimorphos boulder sizes between 1 and 5 m show a clear departure from the power-law fit, which is not due to a resolution effect. Moreover, if we consider the full 1-16 m size range we identify that the best fitting curve to the data is the Weibull function. This function, which is in the form of $y = A \exp\left(-\left(\frac{x}{\lambda}\right)^k\right)$, i.e. the upper cumulative distribution function of the Weibull distribution, it is characterized by the so-called shape parameter $k > 0$ (which affects the shape of the distribution) and by the scale parameter $\lambda > 0$ (which stretches or shrinks the distribution). In the specific case where $k = 1$, the Weibull distribution becomes the exponential one. This function is largely used in fracture and fragmentation theory^{52,51,46,53}, well describing the particle distribution that is derived from grinding experiments⁵⁴”.

+ Fig 4. Same remark as for Fig 1.

Reply: Following the Reviewer’s suggestion, as for Fig. 1A, B, C we have changed Fig. 4A, B and C, better highlighting the study area with the blue polygon, the ellipses with a pink, thicker outline, as well as making the coordinates bigger. We hope that the figure is much more readable now. The plot in Fig. 4 C is also bigger and it is of much higher resolution.

+ Fig 3 and 6 do not show remarkable information, and might be dropped in Suppl. material, especially because Table 1 provides about the same information.

Reply: The Referee is right, the same information appears both in Fig. 3 and 6 and Table 1. Since we would prefer to keep both Fig. 3 and 6, we have eliminated Table 1, whose details and values are all discussed in the main text.

+ The power-law indexes are given for the NEAs observed previously, what about the Weibull function for those asteroids?

Reply: This is a good suggestion, thanks for pointing this out. Recent works by ref⁴⁵ on Ryugu, and by ref¹⁴ on Bennu, started to made use of the Weibull function in order to fit their boulder SFD. In the first case, ref⁴⁵ (Figure 2) showed that the power-law fit is a good match for boulders larger than 1-2 m, but if the full cm- to decameter-size range is considered, then the particle SFD is better described by the Weibull curve, especially for boulder-cobbles with sizes below 1 m. On Bennu, ref¹⁴ applied the same Weibull fit

of Ryugu (Figure 15), showing that this could also be suitable to represent the cm- to decameter-size range SFD. Nevertheless, such curve indicated some under-estimation for sizes >20 m, while it partially over-estimated the boulders in the 0.5-8.0 m size range. On the contrary, the power-law fit showed to be better representative of the overall SFD, as is the case for all remaining NEAs Eros⁹, Toutatis¹¹ and Itokawa¹⁰ (for the latter multiple power-law fitting curves¹⁵ have been introduced to explain the different boulder trends observed on the surface).

In order to follow the Reviewer suggestion, we have added all the above text inside the first lines of the Discussion.

+ "Laboratory impact experiments performed with different projectile velocities, target shapes, compositions and strengths have shown..." for which target sizes? In laboratories this must be about small size objects. Can this really be extrapolated to boulder sizes above 1 m (and much bigger).

Reply: Laboratory impact experiments always employ pebbles and cobbles on a scale ranging from millimeters to centimeters. Despite being considerably smaller than the size of boulders identified, they have proved to be extremely useful when their mean axial ratio is compared to the apparent mean axial ratio derived from impact-generated boulders observed on celestial bodies like Eros, Itokawa, and Ryugu^{16,48}. For this reason, we opted to conduct a similar comparison using our Didymos and Dimorphos data, as done by ref^{16,49}. In order to explain this point also in the main text, we have added a footnote where the laboratory impact experiments are mentioned. In addition, we enlarged the part regarding the discussion of the axial ratio to better explain the comparison with laboratory experiments.

+ "The fact that both the primary and the secondary power-law fits reasonably overlap within error bars (Fig. 7) suggests that Dimorphos' surface has directly inherited part of Didymos' boulders." This affirmation and conclusion do not seem obvious. Could this be expanded? How this can be quantified? Especially if there is some segregation in the boulders mass shedding which should then deplete one of the component with respect to the other and change the size distribution between the two asteroids?

This seems to be contradictory as the two distributions are declared similar? And against the main conclusion?

Reply: We thank the Reviewer for this extremely important comment and the multiple related questions, because they all allow us to better express our interpretation.

Firstly, we modified the Introduction to better introduce the topic adding the following lines at the end of the Introduction section: "... One formation hypothesis of such binary bodies is that due to the Yarkovsky–O'Keefe–Radzievskii–Paddack (YORP³³) effect, a larger primary might have experienced continuous spin-up to reach its spin limit. As a consequence, a mass shedding event or fission of some fraction of its body^{26,34,35} occurred. Ejected materials from the primary are predicted to remain in orbit within the system and reaccumulate outside the Roche limit into a small satellite. If the formation of Dimorphos is related to the top shape and rapid spin-up of Didymos by the YORP effect^{26,34,36,37}, it is expected that its boulders previously belonged to the equatorial region of the primary and have a comparable, inherited SFD. Moreover, if this interpretation holds true, we could also expect some sort of equatorial block depletion on Didymos, as a result of the spin-up process, followed by the mass-shedding event. Here we quantitatively test ..."

To provide a better interpretation, as mentioned in the main manuscript, let us begin by assuming that the primary body is characterized by a boulder SFD inherited by the

catastrophic disruption of the parent body. At some point, due to the rapid spin-up caused by the YORP effect, the spin of Didymos reaches the critic value of 2.25 hr, hence triggering surface landslides and mass shedding. Due to the Coriolis effect mentioned in the main text, larger boulders tend to acquire larger kinetic energy and slide faster, but all surface boulders are affected by such mass movements. The exact size limit below which there is a general block number decrease has not yet been feasibly modeled. Nevertheless, we point out that it is not only a specific size that is being shedded from the asteroid, rather all sizes exceeding a minimum limit (not yet modelled). If this interpretation is correct, we should identify a boulder SFD on the secondary asteroid which is similar to the one of the primary, at least for the bigger sizes. On the contrary, since there should be a size limit during the mass shedding event, we suggest a possible boulder diminishment (a deviation from the original fitting curve) towards the smallest sizes. This is what we can see by the deviation from the Dimorphos power-law fitting curve that turns into a Weibull one for the sizes 1-5 m. The identification of a similar boulder SFD (within error bars) for both the primary and the secondary hence suggest that above a specific size limit, all sizes are proportionally being shedded from Didymos, forming the satellite Dimorphos. This is the reason why we affirm that “The fact that both the primary and the secondary power-law fits reasonably overlap within error bars (Fig. 7) suggests that Dimorphos’ surface has directly inherited part of Didymos’ boulders”.

+ "...this mechanism would preferentially eject larger boulders from the surface." That's interesting but that seems counter-intuitive. Are we talking about boulders here or smaller particles? In any case that would produce a different size distribution on both bodies?
(Important note: this comment has been moved above the next one because it is strictly related to the previous one) Reply: We are talking about boulders here, nevertheless, as mentioned in the main text the exact size limit below which there is a general block number decrease has not yet been feasibly modeled. The important point with this process is that there is not only a specific size that is being ejected, rather a group of boulders that exceeds the minimum size limit. This means that if the pre-depleted surface is characterized by a specific SFD, this could be representative also of the shedded material (above the specific minimum size limit). This is why it is important to discover that both Didymos and Dimorphos, although at different size ranges, show a similar SFD.

+ "...even if Didymos current spin is 2.26 hr only a slightly shorter spin period of 2.2596 hr is critical for the primary to initiate surface landslides and mass shedding." Long sentence not very clear, please precise.
Reply: Following the Reviewer’s suggestion we have changed the long sentence that now states: “The Didymos spin is currently 2.26 hr²³. Nevertheless, as showed by ref⁵⁵ only a slightly shorter spin period of 2.2596 hr could trigger surface landslides and mass shedding from the primary³⁰.”

+ The spin rate is much longer  .. is much slower . Cite the rate.
Reply: We have changed the text accordingly and we have added the spin rate of 11.92 h. The sentence now states “On the contrary, for Dimorphos case the spin rate is much slower²³ (11.92 h) than the one of the primary.”

Reviewer #3 (Remarks to the Author):

General aspects

The topic of the manuscript is to provide new observational based evidence or strong indication on the formation process of an asteroid satellite: Dimorphos. The topic is important and relevant, not only from scientific point of view but also helps planning asteroid impact mitigation actions with the better understanding of asteroid interiors, surface processes and dynamic evolution of these solid bodies. The methods are moderately well described, the language is very good, the illustrations are useful and the references are relevant. There is new result presented, and the audience is waiting for such publication as it presents an important outcome of the recent DART mission. However, some moderate improvements are still needed to make the work publishable, thus the referee suggests moderate revise.

Reply: We thank the Reviewer #3 for the kind words and the important comments, suggestions and corrections received. We hereafter reply to all points in bold.

Several literature sources are indicated why a specific boulder distribution etc. indicates former disruption or other processes. Some further explanation (1-2 sentences) would help many readers to get the related specific information here.

Reply: We thank the Reviewer for the suggestion, we added the following sentence to help the reader to get the related specific information as follows: “In particular, the investigation of block SFDs obtained from size range between few centimeters to hundreds of meters, were found to commonly follow power-law fits^{9,10,11,13,14,15}. From a formative perspective, this means that these boulders have been generated by a sudden fragmentation, as an impact event, and leading to a distribution of remnants characterized by fractals^{8,14}”.

Specific aspects

around 90-95 lines

it would be useful to indicate the maximal boulder size for each mentioned bodies

Reply: Thanks for the suggestion. The maximum boulder size for each mentioned body is indicated in the size-ranges where the SFDs have been derived. We have added one line explaining this in the text. For the case of (162173) Ryugu it is Otohime Saxum with a size of 160 m (ref¹³). On (101955) Bennu there is a boulder with a size range of 90-100 m located in the southern hemisphere of the NEA (ref⁶). On (25143) Itokawa the largest boulder is 50 m wide (ref^{10,12}), for (433) Eros it is 140 m (ref⁹), while for (4179) Toutatis is 61 m (ref¹¹).

108

“relatively small secondaries”

not clear what these „small secondaries” mean

Reply: Within the NEA binaries group, the largest sub-group is the one that is characterized by secondary/primary size ratios of $0.1 \leq \frac{\text{secondary size}}{\text{primary size}} \leq 0.6$ (ref³²). Since the “relatively small secondaries” sentence was not clear nor quantitative, we eliminated it. The sentence now states: “...belongs to the largest group of NEA binaries

with secondary/primary size ratios³² of $0.1 \leq \frac{\text{secondary size}}{\text{primary size}} \leq 0.6$.”

110

„Paddack effect (YORP33),”
shift the „effect” after the bracket

Reply: Done.

120

at the end of the paragraph some expectations could be mentioned also

Reply: Following the Reviewer’s suggestion we have changed and added some new lines inside the paragraph that now states: “If the formation of Dimorphos is related to the top shape and rapid spin-up of Didymos by the YORP effect^{26,34,36,37}, it is expected that its boulders previously belonged to the equatorial region of the primary and have a comparable, inherited SFD. Moreover, if this interpretation holds true, we could also expect some sort of equatorial block depletion on Didymos, as a result of the spin-up process, followed by the mass-shedding event”.

152

“yellow dot”

there are still persons who print the papers with black and white to read, thus suggest to make the colour coded features visible in B&W prints

Reply: Following the Referee’s suggestion, we have changed the yellow dot into a white triangle that is visible also in B&W prints.

166

“NEA power-law”

what is the „NEA” for?

Reply: NEA stands for Near-Earth Asteroids. Since this was unclear, we changed the sentence that now states “... to compare our SFD with power-law fitting curves available from the Near-Earth Asteroid (NEA) literature ...”

207

“The Didymos surface”

it would sound better „the surface of Didymos”

Reply: Done.

Figure 5

I would expect a bit more discussion on the diagram on what could be learned from it

Reply: Thanks for the important suggestion. Inside the Results – Dimorphos and Results – Didymos sections we have tried to be as synthetic as possible by presenting the main results obtained but not discussing them there. This was done to avoid any possible repetition with the Discussion part of the manuscript. On the contrary, inside the *Discussion – Evidence for formation of Dimorphos via mass shedding* there are 16 lines discussing the plots presented in Fig. 2 and 5 that are hereafter copied: “The Dimorphos formation scenario via mass wasting is also supported by plotting the size of its boulders versus latitude, longitude, slope, gravitational acceleration and potential (Fig. 2A-E). Indeed, we found that the boulders appear to be randomly distributed on the surface. Nevertheless, the clear cut-off of boulders located on gravitational slopes in the 35°- 45° range suggests that no blocks can remain stable at larger inclinations³⁰. Consequently, this implies that the angle of repose, denoting the maximum angle at which granular

material can be piled without collapsing, for Dimorphos's material falls within this specific range. Contrarily, on Didymos the size versus latitude plot (Fig. 5A) suggests that the largest sizes are more concentrated at the highest latitudes where the rough highland is located³⁰, i.e. further from the equatorial triangular-shaped ridge. This supports the interpretation that the equatorial “smooth” lowland of Didymos³⁰ is characterized by boulders with sizes that are close to or under the DRACO detection limit. This might be the result of the mass-shedding event that generated Dimorphos from Didymos equatorial band⁵⁵, which later flattened, being characterized only by small rubbles. The boulder size versus longitude, slope, gravitational acceleration and potential (Fig. 5B-E) do not exhibit a specific trend indicating a random distribution, as observed for Dimorphos. Nevertheless, a gravitational slope boulder cut off in the 55°-65° range suggests that on Didymos there is surface cohesion which is larger than Dimorphos’s one”.

If the Referee agrees, we would like to keep the text separated. Nevertheless, we now mention in Fig. 2 and 5 captions that a detailed discussion of such diagrams and their implications for formative and degradation processes is presented in the *Discussion - Evidence for formation of Dimorphos via mass shedding* section.

229

“previously studied NEA global boulder SFDs”

is NEA for Near Earth Asteroids? Resolve the acronym. And would be good a short sentence on the main characteristics of these SFDs.

Reply: Yes, this is correct. We have changed the sentence as “The previously studied global boulder SFDs derived from Near-Earth Asteroids...”.

Regarding the short sentence on the main characteristics of the SFDs, we describe what a power-law fitting curve means from a formative perspective two lines below. In addition, we discuss the main SFDs characteristics right afterward. If the Referee agrees, we would like to keep such explanation in this part of the Discussion section.

236

„respectively9,10,11,16.”

suggest to finish the sentence like this: „respectively 9,10,11,16 thus represent...” and continue the argumentation

Reply: Good suggestion. We have changed the sentence that now states: “As shown in Fig. 7B, boulders on all stony NEAs previously visited by spacecrafts—Itokawa, Eros and Toutatis—are characterized by power-law fitting curves with indices steeper than -3.0. In particular, Itokawa shows a power-law index of -3.05 ± 0.14 for boulders $\geq 5 \text{ m}^{10}$, Eros has a power-law index of -3.25 ± 0.14 for boulders $\geq 10 \text{ m}^9$, while Toutatis is characterized by a power-law index of -4.4 ± 0.1 for boulders $\geq 20 \text{ m}^{11}$. On the contrary, on carbonaceous asteroids Ryugu and Bennu the power-law indexes obtained are -2.65 ± 0.05 for boulders $\geq 5 \text{ m}^{13}$ and -2.5 ± 0.1 for boulders with sizes $\geq 0.2 \text{ m}^{14}$, respectively. Such indices all confirm an impact-related formation that led to an SFD characterized by fractals⁴⁶.

239 „indicate an impact-related origin”

and

242 „catastrophic disruption”

not straightforward for all readers why do these aspects „imply”, suggest to briefly explain

Reply: In the previous suggestion for line 236 we have expanded the sentence mentioning that a power-law fitting curve confirms an impact-related formation which leads to an SFD characterized by fractals (ref⁴⁶). We believe that this should clarify the

points raised for lines 239 and 242. In addition, we slightly changed the following lines in order, we hope, to be more straightforward for all readers. The text now states “**The Dimorphos power-law index α of -3.4 ± 1.3 , obtained for boulders ≥ 5 m, and Didymos power-law index α of -3.6 ± 0.7 , derived from boulder sizes ≥ 22.8 m, confirm this generally steeper stony boulder SFD (Fig. 7B) when compared to the carbonaceous one, as well as they indicate a sudden, impact-related origin for the identified boulders^{10,11,16}. As for the other visited bodies, this evidence, coupled with the maximum identified boulder dimensions (93 m on Didymos and 16 m on Dimorphos) that both exceed 1/10 the NEAs’ diameters, imply that such asteroids are collections of debris resulting from the catastrophic breakup of a larger parent body^{16,8,30,31}, followed by the reaccretion of part of its fragments”.**

256

„during granular processes”

suggest to cite: <https://ui.adsabs.harvard.edu/abs/2014P%26SS..101...65K/abstract>

Reply: Great suggestion! Added to the manuscript. We have changed all References’ numbers accordingly.

Figure 7

suggest to make such line pattern coding (beside the colour) that allows to separate these curves on black and white prints also

Reply: Good idea. We have changed Fig. 7 accordingly.

286

„suggests that Dimorphos’ surface has directly inherited part of Didymos’ boulders”

OK, but indicate why are there smaller boulders on Dimorphos

Reply: The fact that both the primary and the secondary have similar power-law fits (in particular Dimorphos largest boulders) suggest the mass inheritance of Dimorphos from Didymos, but this does not mean that during the mass shedding event only “large” boulders have been ejected. Instead, also the smaller component could have been shed and deposited on Dimorphos, as we could, for example, identify in the impact site with DART/DRACO 0.05-m spatial scale images (see the following image we prepared for an Impact-site study related manuscript lead by a DART colleague - in preparation). The fact that smaller (<5 m) boulders are present on Dimorphos but show a Weibull-distribution best fit is then discussed in the manuscript (presented in the three Discussion sections).

Boulder-cobble identification in the DART impact site location. The image used is the last, full DRACO 0.05 m spatial scale one before the spacecraft disintegration.

305-306

at the „clear cut-off”

some further explanation on what does it mean and indicate would help

Reply: We have expanded the sentence that now states “Nevertheless, the clear cut-off of boulders located on gravitational slopes in the 35°- 45° range suggests that no blocks can remain stable at larger inclinations³⁰. Consequently, this implies that the angle of repose, denoting the maximum angle at which granular material can be piled without collapsing, for Dimorphos's material falls within this specific range”.

317-319

not very clear sentence, suggest to reformulate

Reply: We have reformulated the sentence that now states “The alignment of Dimorphos's boulders' semi-major axes relative to the local north could offer insights into their

potential source location on Didymos. Indeed, if these boulders exhibit random orientations, it would imply an accumulation of boulders sourced from multiple locations on Didymos”.

320

„from random locations from Didymos”

this sentence is not clear enough, do the authors mean boulder transport from Didymos to Dymorphos much after their formation?

Reply: We have rephrased and changed the previous sentence from lines 317-320. Here we are talking about the mass shedding event that generated Dimorphos, not the boulder transport occurring much after their formation. We hope that with the new sentence it is much clearer now.

333

„the gravitation is”

suggest to modify to „the gravity”

and change the „smaller” to „weaker”

Reply: Good suggestion. Done.

Figure 8

to better understand and interpret the orientations, the relation of orientations on the two bodies relatively to each other (Dimorphos is tidally locked) should be indicated

Reply: We have changed the caption of Fig. 8 following the Reviewer’s suggestion. It now states: “Fig. 8. A: Rose diagrams of Dimorphos (A) boulders ≥ 1 m and of Didymos (B) boulders ≥ 16.5 m with an apparent axial ratio < 0.9 . The corresponding mean orientation and standard deviation are also indicated. We recall that Dimorphos is tidally locked with respect to Didymos, and the hemisphere of the secondary observed by DRACO is approximately perpendicular to the Didymos facing-side⁴⁴”.

Table 2

enlarge the text size

Reply: Done.

398

„belongs to the major binary NEA group”

what does that mean? What is the major group? Or do you mean „average object make up most of binary NEAs”?

Reply: The Referee is correct, the sentence was not clear. We have changed it as “Since Didymos is part of the largest group of binary Near-Earth Asteroids with a secondary-to-primary size ratio ranging from 0.1 to 0.6, the presented results give insights into the formation of such secondaries as a consequence of boulder shedding from the primary asteroid. Moreover, this contextualization of binary NEAs within the broader framework of small bodies enhances our understanding of their formation mechanisms and contributes to a more comprehensive perspective on the dynamics of such systems.”

Please indicate the possible errors of the boulder size measurement in the Methods section

Reply: Inside the Methods section there are now five new lines about the boulder size identification and the possible error. The new sentence now states “Since we are identifying all boulders larger than three pixels, the associated size-identification error does not exceed one pixel, as showed in ref^{14,39}, with the corresponding SFD falling

within the uncertainty¹⁴. Nevertheless, as mentioned in the main text, in order to be conservative on the counts and prevent potential size misinterpretation, we opted to raise the minimum size threshold deemed reliable from three to five pixels”.

It would be better to have somewhere a high-resolution image example on “nice” boulders on Dimorphos

Reply: Thanks for the suggestion. We have prepared three closeup images in order to show both the uninterpreted boulders, as well as the identified ones. We have added such three images inside the Supplementary Information. Inside the Methods we have added two lines stating: “Three closeup images showing the uninterpreted surface and the boulders identified with pink ellipses have been added into the Supplementary Information”.

REVIEWERS' COMMENTS

Reviewer #1 (Remarks to the Author):

The authors have carefully addressed the feedback provided, implementing relevant changes to enhance the manuscript. Following a thorough review of the revised document, I believe it is acceptable for publication.

best regards,
Reviewer

Reviewer #2 (Remarks to the Author):

The authors have replied in detail to most of my questions and have improved the manuscript in several ways making the paper more clear about the results and their interpretation. I recommend the publication without further corrections.

Reviewer #3 (Remarks to the Author):

Review of Evidence for multi-fragmentation and mass shedding...

General aspects

The authors revised their first submission, what made the work much better. The referee thinks the manuscript is almost ready for publication, some minor modifications are suggested below.

Specific aspects

“when compared to the carbonaceous one”

not clear enough what is the carbonaceous case here and what is its relevance, regarding asteroid (and not meteorite) classification, C-type is more often used

“indicate a sudden, impact-related origin”

what is „sudden for”? not clear what is provided as further info beyond the „impact origin”

„underscores the distinct responses of materials (stony versus carbonaceous) to meteoroid impacts and thermal cracking”

important, might mention in the abstract too

“the density per km² of Dimorphos boulders ≥ 1 m is 2.3x with respect to the one obtained for Bennu, while it is 3.0x with respect to Ryugu. Such values increase once Dimorphos boulders ≥ 5 m are compared with Bennu (3.5x), Ryugu (3.9x) and Itokawa (5.1x).”

this part is quite useful please mention in the abstract also, however this statement also there

“binary systems are affected” is too strong as this is the only one such already visited system, so suggest to reformulate it

„on carbonaceous asteroids”

again, C-type asteroids is more relevant term

Reply to Reviewer 3

Dear Reviewer, thanks a lot for your kind words about the Revision. We hereafter reply to the final points raised.

“when compared to the carbonaceous one”

not clear enough what is the carbonaceous case here and what is its relevance, regarding asteroid (and not meteorite) classification, C-type is more often used

Reply: This is correct. Nevertheless, since Ryugu is a C-type, but Bennu is a B-type (both belonging to the C-complex/carbonaceous asteroids group, but we cannot affirm that Bennu is a C-type object) we would prefer to keep the word *carbonaceous* in the main text. Nevertheless, in order to be clear on this aspect, we have now changed the following sentence for both C-complex and S-complex asteroids.

The sentence now states: “The power-law indexes obtained on global counts for C-complex asteroids (henceforth called carbonaceous asteroids) (162173) Ryugu and (101955) Bennu are -2.65 ± 0.05 for boulders..... . On the other hand, the global boulder distribution of the S-complex (henceforth called stony) NEA (25143) Itokawa has a power-law index of....”.

“indicate a sudden, impact-related origin”

what is „sudden for”? not clear what is provided as further info beyond the „impact origin”

Reply: This is right, thanks for the suggestion. We have eliminated the word sudden.

„underscores the distinct responses of materials (stony versus carbonaceous) to meteoroid impacts and thermal cracking”

important, might mention in the abstract too

Reply: This is a good suggestion, thanks. Nevertheless, since we are not mentioning the carbonaceous (C-complex) asteroids in the abstract versus the stony (S-complex) objects we would like to avoid inserting this sentence right at the beginning of the manuscript, if the Referee agrees.

“the density per km² of Dimorphos boulders ≥ 1 m is 2.3x with respect to the one obtained for Bennu, while it is 3.0x with respect to Ryugu. Such values increase once Dimorphos boulders ≥ 5 m are compared with Bennu (3.5x), Ryugu (3.9x) and Itokawa (5.1x).”

this part is quite useful please mention in the abstract also, however this statement also there “binary systems are affected” is too strong as this is the only one such already visited system, so suggest to reformulate it.

Reply: Following the Referee’s suggestion we have now changed the sentence that states: “it could mean that contrarily to the single bodies visited so far, binary systems might be affected by subsequential fragmentation processes that largely increase their block density per km²”.

In addition, we have added the following lines in the abstract following the Reviewer’s suggestion: “The density per km² of Dimorphos boulders ≥ 1 m is 2.3x with respect to the one obtained for (101955) Bennu, while it is 3.0x with respect to (162173) Ryugu. Such values increase once Dimorphos boulders ≥ 5 m are compared with Bennu (3.5x), Ryugu (3.9x) and (25143) Itokawa (5.1x). This is of interest in the context of asteroid studies because it means that contrarily to the single bodies visited so far, binary systems might be affected by subsequential fragmentation processes that largely increase their block density per km²”.

„on carbonaceous asteroids”

again, C-type asteroids is more relevant term

Reply: See the Reply to the first point raised above.